# Seasonal ice storage changes and meltwater generation at Murtèl rock glacier (Engadine, eastern Swiss Alps): Estimates from measurements and energy budgets in the coarse blocky active layer

Dominik Amschwand[1], Seraina Tschan[1], Martin Scherler[1,†], Martin Hoelzle[1], Bernhard Krummenacher[2], Anna Haberkorn[2], Christian Kienholz[2], Lukas Aschwanden[3], and Hansueli Gubler[4]

[1]Department of Geosciences, University of Fribourg, Fribourg, Switzerland
[2]GEOTEST AG, Zollikofen/Bern, Switzerland
[3]Institute for Geological Sciences, University of Bern, Bern, Switzerland
[4]Alpug GmbH, Davos, Switzerland
[†]deceased, 4 June 2022

**Correspondence:** Dominik Amschwand (dominik.amschwand@unifr.ch)

**Abstract.** Intact rock glaciers, a permafrost landform common in high-mountain regions, are often conceptualized as (frozen) water reserves. In a warming climate with slowly degrading permafrost, the large below-ground ice volumes might suggest a buffering effect on summer streamflow that due to the climate resiliency of rock glaciers only increases with rapidly receding glaciers. In this case study, we assess the role and functioning of the active Murtèl rock glacier in the hydrological cycle of

its small (30 ha) periglacial and unglacierized watershed located in the Upper Engadine (eastern Swiss Alps). Our unprecedentedly comprehensive hydro-meteorological measurements include below-ground heat flux measurements in the $3-5\,\mathrm{m}$ thick coarse-blocky active layer (AL), direct observations of the seasonal evolution of the ground-ice table, and discharge and isotopic signature of the outflow at the rock-glacier front. The detailed active-layer energy and water/ice balance quantifies precipitation, evaporation, snowmelt, ground ice melt, and catchment surface outflow. Our single-site, but detailed case study

resolves thermo-hydraulic processes in the coarse blocky AL that might enhance the snowmelt–groundwater connectivity in periglacial high-mountain watersheds underlain by discontinuous permafrost. A substantial part of the snowmelt refreezes in the cold AL ($\sim 150-300\,\mathrm{mm}$ w.e. or $\sim 20-40\,\%$ of the snowpack), forming superimposed AL ice that is released during the thaw season at melt rates low enough for the meltwater flow to be routed through the permafrost aquitard to deeper sub-permafrost aquifers. Meltwater fluxes are low ($1-4\,\mathrm{mm}$ w.e. $\mathrm{day}^{-1}$), but sustained throughout the entire thaw season

($\sim 100\,\mathrm{days}$) due to small ground heat fluxes and the dampening effect of the AL. The superimposed AL ice acts as a coupled thermo-hydrological buffer that (to some extent) protects the underlying ice-rich rock glacier core by converting most of the ground heat flux to meltwater during the thaw season. Consequently, meltwater release from the 'old' permafrost ice due to climate-induced permafrost degradation is currently $\sim 10\,\mathrm{mm}\,\mathrm{yr}^{-1}$ or an order of magnitude smaller than the contribution of AL meltwater and not more than a few $\%$ of the overall water/ice fluxes. In view of the widespread and long-lasting oc-

currence of climate-resilient permafrost in high-mountain watersheds and the increasing importance of groundwater-sustained late-summer baseflow relative to vanishing glaciers and diminishing snowpacks, it is important to investigate mechanisms, flow paths, and efficiency of groundwater recharge in mountain permafrost terrain.

# 1   Introduction

In our rapidly changing climate, changing precipitation patterns and enhanced sublimation/evapotranspiration due to warming and greening (vegetation succession) lead to profound hydrological regime shifts in high-mountain regions (Hock et al., 2022). The hydrological buffer capacity of *above-ground* cryospheric components decreases, as glaciers recede and precipitation in form of snow decreases, while evaporative losses to the atmosphere (Fugger et al., 2024) and inter-annual variability in precipitation increase. Droughts and reduced stream flow in the summer months of dry years might become more frequent and severe, reducing water security in downstream regions (Haeberli et al., 2017; Schaffer et al., 2019; Hoelzle et al., 2020; Barandun et al., 2020; Arenson et al., 2022). Hence, the hydrological buffer capacity of comparatively climate-resilient and robust *below-ground* components will become increasingly important: ice-rich mountain permafrost with rock glaciers as its most conspicuous morphological expression (Barsch, 1996), and groundwater in headwater aquifers (Woo, 2011; Hayashi, 2020).

Intact (i.e., ice-bearing) rock glaciers, a mountain permafrost landform widespread in nearly all mountain ranges worldwide, are distinct bodies of a perennially frozen debris–ice mixture covered by a seasonally thawed debris layer, the active layer (AL) (Haeberli et al., 2006). The perennially frozen interior, the rock glacier core, consists of ice-supersaturated debris whose creep deformation results in the conspicuous lobate or tongue-like form (Barsch, 1996). Intact rock glaciers store and release water in different forms, such as ice, snow, and water, on long-, intermediate- and short-term timescales (Jones et al., 2019). The insulating and convective cooling effect of the thick AL creates a cool and stable microclimate in the AL that is partially decoupled from the atmosphere (Wakonigg, 1996; Humlum, 1997; Harris and Pedersen, 1998; Hanson and Hoelzle, 2004; Delaloye and Lambiel, 2005; Guodong et al., 2007; Amschwand et al., 2024a, b). Controlled by the ground thermal regime rather than by the surface energy balance (SEB) directly (Amschwand et al., 2024a), ground ice melt at depth proceeds slower and delayed compared to snow or glacier melt at the surface on seasonal up to decadal timescales (cf. Reznichenko et al., 2010). On a seasonal time scale, the sustained melt of ground ice is thought to contribute to late-summer streamflow. On decadal time scale, rock glaciers are more resilient (less sensitive) to climate changes compared to (debris-covered) glaciers and expected to outlast them as our mountains shift away from the glacial towards the para- and periglacial realm (Haeberli et al., 2017, 2024; Schaffer and MacDonell, 2022). Especially in semi-arid, already today weakly glaciated and water-stressed high mountain areas such as parts of Central Asia, the Himalayas or the Dry Andes, the large below-ground ice volumes and hydrological buffer capacity of climate-resilient, ice-rich permafrost landforms might become important hydrological elements.

The hydrological significance of rock glaciers primarily relates to (1) the long-term, climate-resilient storage of permafrost ice in the rock glacier core (water in reserve), (2) the seasonal storage and freeze/melt of water/ice in the AL (water/ice in circulation), and (3) water storage in unfrozen fine-grained sediments and interaction with liquid water flowing through or beneath rock glaciers (storage–release, routing and chemical alteration/mineralization of water) (Azócar and Brenning, 2010; Corte, 1976; Burger et al., 1999; Jones et al., 2019). An useful concept for their understanding and management is to differentiate between water/ice in circulation (renewable, flow-limited resource) and water/ice in reserve (nonrenewable, stock-limited resource) based on a time scale of one (or a few) hydrological years (Gleick and Palaniappan, 2010).

Intact rock glaciers have been storing 'old' ice in the permafrost body beneath the AL for centuries up to millennia (Krainer et al., 2015). This permafrost ice at depth has been interpreted to be roughly as old as the rock glacier it is embedded in. Therefore, it is frozen precipitation from past Holocene cold climatic phases and is a nonrenewable resource (Barsch, 1996;

Haeberli et al., 2003; Azócar and Brenning, 2010; Krainer et al., 2015; Amschwand et al., 2021; Lehmann et al., 2022; Nickus et al., 2023). Protected by the AL, accumulation and melt have been slow and driven by major climatic shifts throughout the Holocene. Driven by the current climate change, permafrost including rock glaciers is found to be degrading widely in high-mountain environments (Beniston et al., 2018; Biskaborn et al., 2019). Intact rock glaciers react by slow AL deepening, releasing meltwater previously bound in the ice-rich permafrost, and transitioning towards a relict (i.e., ice-free) state over the

time scale of centuries (Scherler et al., 2013). As intact rock glaciers are a common periglacial landform and the ice-rich core is typically $10-30$ m thick (Cicoira et al., 2021), the amount of below-ground permafrost ice as estimated from (static) rock glacier inventories and empirical area-volume scaling relations is substantial (Rangecroft et al., 2015; Azócar and Brenning, 2010; Brenning and Azócar, 2010; Brenning, 2005; Jones et al., 2018b, a). In semi-arid, weakly glacierized catchments, water volume equivalent (w.e.) stored in rock glaciers can exceed glacier ice volume or might do so in the future (Jones et al., 2018a;

Bodin et al., 2010; Janke et al., 2017; Azócar and Brenning, 2010).

'Young' ice accumulates and melts seasonally in the AL ("seasonal AL ice", "superimposed ice" in Bearzot et al. (2023)). This ice is derived from modern precipitation and is a renewable resource. Its accumulation/release is conditioned by freeze/thaw cycles driven by the SEB and short-term weather conditions. AL ice volumes are presumably much smaller compared to that of permafrost ice, but in much faster exchange between the atmo- and hydrosphere. Available studies suggest that a considerable

fraction of the annual precipitation can be stored and released from intact rock glaciers. Marchenko et al. (2012) report that the observed accumulation–melt of $40-60$ cm of ground ice seasonally stored a substantial amount ($< 30\%$) of the snowpack in the Northern Tien Shan. Halla et al. (2021) found inter-annual ice storage changes on Dos Lenguas rock glacier (Dry Andes of Argentina) of $-36$ mm yr$^{-1}$ ($25-80\%$ of the annual precipitation) and $+28$ mm yr$^{-1}$ ($17-55\%$). While the timing of seasonal formation and melt of ground ice has long been inferred from thermal (Hinkel and Outcalt, 1994; Kane et al., 2001;

Hanson and Hoelzle, 2003, 2004; Sawada et al., 2003; Rist and Phillips, 2005; Herz, 2006) and geophysical measurements (Hilbich et al., 2009; Schneider et al., 2013), so far few estimates of the magnitude of seasonal to inter-annual ice turnover exist even at plot scale. These estimates are based on rare exposure of ground ice or drillings (Sawada et al., 2003; Yoshikawa et al., 2023; Marchenko et al., 2012, 2024), and petrophysical joint inversions of (most often) geoelectrical and seismic measurements (Mollaret et al., 2019, 2020; Steiner et al., 2021; Bearzot et al., 2023; Bast et al., 2024), in cases combined with

kinematic surveys to relate changes in ground ice content to surface kinematics (heave/subsidence) (Halla et al., 2021). Quantifying changes in ground ice content and AL depth is a progressing research field (e.g. Hauck et al., 2011; Wagner et al., 2019; Pavoni et al., 2023; Maierhofer et al., 2024; Morard et al., 2024) because the closer degrading permafrost approaches $0\,°C$ and enters the zero curtain, the more its state is reflected by changes in ice content (latent heat) rather than by ground temperatures (sensible heat) (Hauck and Hilbich, 2024). Other approaches include hydro-chemical and water isotope measurements (Blum-

stengel and Harris, 1988; Harris et al., 1994; Williams et al., 2006; Munroe and Handwerger, 2023a, b), AL energy budgets (Scherler et al., 2014), and numerical modelling (Pruessner et al., 2022; Renette et al., 2023; Mendoza López et al., 2024).

The ice-rich permafrost of intact rock glaciers affects the hydrology of a watershed by its semi-impervious layer (aquitard) that controls the lateral flow and limits (but not prevents) the exchange between surface waters and sub-permafrost groundwater (Cheng and Jin, 2012). A perched supra-permafrost aquifer within the AL on top of the ground-ice table is separated from a sub-permafrost aquifer (and unfrozen water in intra-permafrost taliks). Runoff from the shallow, highly permeable supra-permafrost aquifer is rapid and flashy, this quickflow is only weakly chemically altered. However, some water can percolate across the permafrost through taliks or 'warm', permeable zones. These are characteristic features of the inherently discontinuous mountain permafrost, whose distribution is mosaic-like and controlled by spatially complex topo-climatic conditions like insolation/shading and snow cover (Arenson et al., 2022). A dynamic sub-permafrost storage of liquid water retains water in fine-grained, unfrozen sediments during months–years and sustains baseflow during summer droughts or in winter (Rogger et al., 2017; Wagner et al., 2021; Reato et al., 2021; Bearzot et al., 2023). This water is more strongly chemically altered and mineralized, as for example indicated by its higher electrical conductivity (Krainer and Mostler, 2002). For long-term projections, it is important to appreciate the widely diverging time scales of storage mechanisms (from weeks in the AL to millennia in the permafrost core) and the transient state of the mountain permafrost subject to climate change (Jones et al., 2019). As rock glaciers degrade, hydraulic permeability and liquid water storage capacity increases at the expense of the storage of permafrost ice (Winkler et al., 2016a, b; Wagner et al., 2016; Harrington et al., 2018; Colombo et al., 2018).

Although the conceptual framework of the different rock glacier storage mechanisms is established knowledge, no consensus on the present nor future hydrological role of rock glaciers has been reached to date (Duguay et al., 2015; Schaffer et al., 2019; Jones et al., 2019; Arenson et al., 2022), not least due to sparse quantitative hydro-meteorological field data from permafrost-underlain high-mountain watersheds.

In this work, we present quantitative data from a detailed single-site case study on Murtèl rock glacier (Engadine, eastern Swiss Alps) and compare short-, intermediate and long-term water/ice storage changes for the hydrological years 2021–2023. We estimate the ground ice storage changes based on point-wise in-situ observations of the seasonally moving ground-ice table and from the AL energy budget. Ground ice storage changes are energy-controlled phase changes, i.e., ice/water and energy turnover in the AL are closely linked. By measuring and parameterizing the ground heat fluxes and accounting for sensible heat storage changes in the thick debris mantle (Amschwand et al., 2024b), the latent storage changes associated with water phase changes — melting and refreezing — can be isolated. We build on a large body of local previous work: The SEB and AL-internal heat fluxes on Murtèl have been measured/estimated by Mittaz et al. (2000); Hoelzle et al. (2001); Stocker-Mittaz et al. (2002); Schneider (2014); Scherler et al. (2014); Hoelzle et al. (2022) and Amschwand et al. (2024a, b). Hydro-meteorological measurements (snow, rainfall, outflow discharge) complement the plot-scale water budget. Additionally, we use stable water isotope and electrical conductivity (EC) to compare the rock-glacier outflow to the known $\delta^{18}$O–EC signature of the permafrost ice. This case study contributes to the question of the hydrological significance of rock glaciers by presenting a complete hydro-meteorological data set at the well-studied Murtèl rock glacier and explores the permafrost–groundwater connectivity.

## 2 Study site

### 2.1 Rock glacier structure and hydro-morphological setting

The studied Murtèl rock glacier (Murtèl I; WGS 84: 46°25′47″N, 9°49′15″E; CH1903+/LV95: 2′783′080, 1′144′820; 2620–2700 m asl.; Fig. 1, A1), close-by Marmugnun rock glacier (Murtèl II) and the relict Murtèl III rock glacier are located in a north-facing cirque in the Upper Engadine, a slightly continental, rain-shadowed high valley in the southeastern Swiss Alps (Fig. 2a). Mean annual air temperature (MAAT) is $-1.7°$C, mean annual precipitation (MAP) is $\sim 900$ mm (Scherler et al., 2014). The rock glaciers have an altitude range from 2540 (base of front Murtèl III), 2620 (base of front Murtèl I) to 2720 m asl (transition to talus) (Fig. 2b). The talus slopes (at an elevation 2720–2800 m asl.) connect the active rock glaciers to the headwalls and consist of large, angular debris. The headwalls rise from 2800 to 3165 m asl. (a spur of Piz Murtèl) and are more active above rock glacier Marmugnun (Müller et al., 2014) with massive, long-lasting avalanche deposits in late spring–early summer and a debris cone built by frequent rock falls in summer–autumn. The entire mountain slope is part of the periglacial belt and underlain by permafrost (Müller et al., 2014). Perennial snow patches/névés reported by Haeberli (1990) and Tenthorey and Gerber (1991) disappeared by the early 2000s (M. Hoelzle, pers. comm.), but were exceptionally present in the cool-wet summer 2021. Soils are absent–thin and vegetation is sparse. The catchment is small (30 ha) and not glacierized.

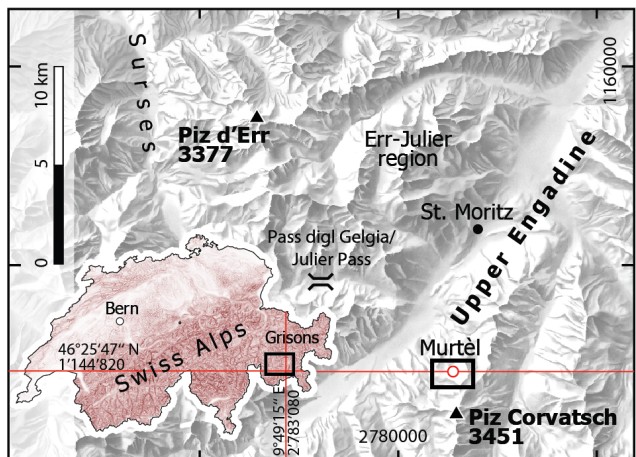

**Figure 1.** Location of Murtèl rock glacier in the Upper Engadine, a high valley in the eastern Swiss Alps. Inset map: Location and extent (black rectangle) of regional map within Switzerland (source: Swiss Federal Office of Topography swisstopo).

The lobate Murtèl rock glacier is ca. 300 m long, 180 m wide, and covered by a $2-5$ m thick coarse-blocky AL (debris mantle). Geophysical investigations revealed that its thickness varies according to the surface micro-topography, from 2 m in the furrows to 5 m beneath the ridges (Vonder Mühll and Klingelé, 1994; Vonder Mühll et al., 2000). Thus, the permafrost table shares the surface furrow-and-ridge micro-topography, although attenuated (water ponding or channelling?). Fine material increases towards the AL base, but is overall sparse. The permeable coarse-blocky AL does not inhibit water flow and has a very low water retention capacity (Springman et al., 2012). The Murtèl permafrost body between the seasonally thawed coarse-

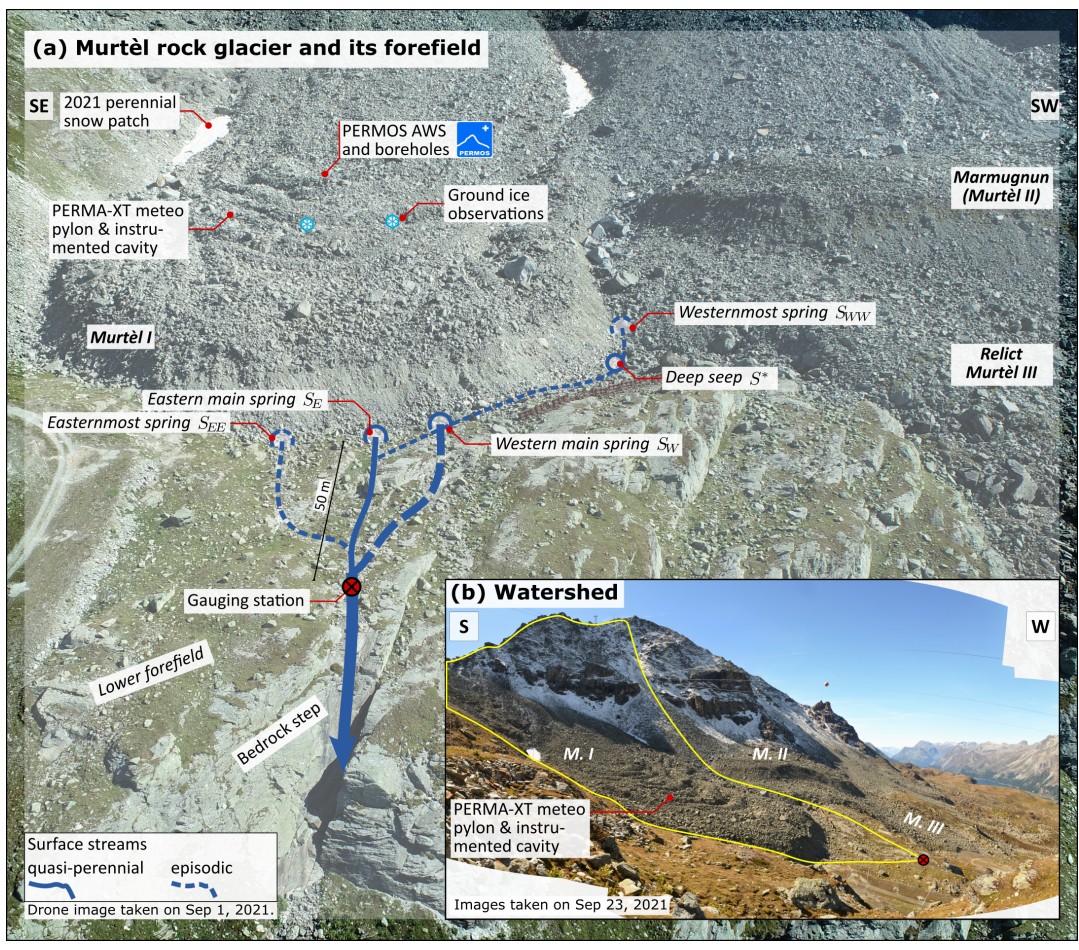

**Figure 2. (a)** Oblique aerial view of the Murtèl rock glacier and its forefield showing location of the rock glacier springs $S_{EE}$, $S_E$, $S_W$, $S_{WW}$ and $S^*$. Parallel rock ledges show a set of N-S running fractures. **(b) Inset** The Murtèl periglacial catchment.

blocky AL (0–3 m) and bedrock (at 50 m) is comprised of three distinct layers (Vonder Mühll and Haeberli, 1990; Haeberli, 1990; Arenson et al., 2002): (1) massive ice, sparsely sand- and silt-bearing (3–28 m, supersaturated with over 90% ice content); (2) a layer of ice-saturated frozen sand (28–32 m) accommodating ca. 60% of the total/surface displacement (shear horizon); and (3) ice-saturated debris (32–50 m; 40% ice). The three deep boreholes (Fig. 2), all located within 30 m distance, share this three-part stratigraphy, but also reveal lateral small-scale material differences (e.g., laterally variable ice/sand content, lenses) and thermal anomalies (e.g., pointing at non-diffusive heat transfer and intra-permafrost water flow (Vonder Mühll, 1992; Arenson et al., 2010)). The extent and ice content of the ice-rich permafrost body is well known from boreholes (Vonder Mühll and Haeberli, 1990; Vonder Mühll, 1992; Vonder Mühll and Holub, 1992; Vonder Mühll and Klingelé, 1994; Haeberli and Vonder Mühll, 1996; Vonder Mühll et al., 2000) and geophysical measurements (electrical resistivity tomography (ERT) and seismic refraction tomography) (Hauck, 2013; Arenson et al., 2010; Mollaret et al., 2019). These data indicate that the

ice-rich permafrost core has an extent of $150 \times 300 \ \mathrm{m}^2$ [C. Hauck, pers. comm.], amounting to a water volume equivalent (WVEQ) of $\sim 1.5 \times 10^6 \ \mathrm{m}^3$. Observations at the surface and during drilling operations, borehole temperature and ERT found evidence for supra-permafrost, (artesian) intra-permafrost, and sub-permafrost water flow (Arenson et al., 2002; Springman et al., 2012). Intra- and sub-permafrost conduits were revealed by water leakage into all drilled boreholes at several depths (Arenson et al., 2002), audible water flow, strong air inflow at the surface, and voids photographed by the borehole camera. Overall, even though Murtèl has a relatively massive, 'clean' ice core compared to other permafrost drill cores (Lazaun, Krainer et al. (2015); Nickus et al. (2023)), the permafrost body is far from being impermeable.

The Murtèl rock glacier sits in a bowl-shaped, glacially overdeepened bedrock depression as shown by boreholes and geophysical soundings/gravimetry (Vonder Mühll and Klingelé, 1994). The rock glacier front has advanced beyond the cirque lip (bedrock sill) onto the forefield that slightly dips away to NNW. Four rock-glacier springs and one seep emerge between coarse blocks at the base of the rock-glacier front (Fig. 2). The forefield is thinly covered by glacial sediments (till veneer, few large boulders), not perennially frozen (from ERT), and vegetated by grasses (Schneider et al., 2013). There are no surface waters upslope of the rock-glacier springs, and no surface water bodies are impounded in the catchment (except episodically during snowmelt or rainstorms). The bedrock appears fractured. For a few days after strong precipitation, water flows out of bedrock fractures in the steep rock face below the forefield (Fig. 2). The bedrock of the Murtèl cirque predominantly consists of granodiorite, whose blocks make up the bulk of the talus slopes and rock glacier coarse-blocky AL. This *Corvatsch granodiorite* unit is separated by a tectonic thrust from a westward-thickening seam/wedge of meta-sedimentary units of the *Rusenna formation* and *Blais radiolarite* consisting of weakly metamorphosed limestone, mica schists, and radiolarite.

## 2.2 "Gray" hydrological and hydro-chemical investigations

We summarize three site-specific past hydrological or water (isotope-)chemical studies which are not easily accessible, namely Haeberli (1990), Tenthorey and Gerber (1991), and Stucki (1995).

Murtèl is one of the worldwide few rock glaciers where the chemical and isotopic signature of the difficult-to-access permafrost ice, the supposed source of the meltwater, is known from analysed drillcores (Haeberli, 1990). This report provides a concise overview of the isotope ($\delta^2$H, $\delta^{18}$O, tritium) and major ion chemistry of the 2/1987 drillcore at depths between $3.34$ and $20.92 \ \mathrm{m}$. The $\delta^{18}$O values of the permafrost ice is in the range of $-16$ to $-13$‰, the deuterium excess in the range of $13$ to $15$‰, and the $\delta^{18}$O–$\delta^2$H relationship is $\delta^2\mathrm{H} = 7.97 \delta^{18}\mathrm{O} + 12.67$‰. The electrical conductivity estimated from the major ion concentration is in the range of $5-30 \ \mathrm{\mu S \ cm}^{-1}$. They interpreted the ground ice as refrozen "diluted groundwater" (non-closed ion balance/cation surplus, pH in the range of $6.3-8.6$) likely derived from winter precipitation/snowmelt (syngenetic permafrost formation).

Tenthorey and Gerber (1991) carried out tracer tests in summer 1989 (napthionate and sodium chloride) and found evidence of two types of water flow, a rapid–channelised (supra-permafrost; $> 120 \ \mathrm{m \ h}^{-1}$) and a slow–diffuse (intra-permafrost?; $< 25 \ \mathrm{m \ h}^{-1}$) flow (Figs. A1, A1). The Murtèl supra-permafrost water drains predominantly to the main springs $S_E$ and $S_W$; a hydrological connection to the deep seep $S^*$ and even to the relict Murtèl III rock glacier (Fig. 2b) exists but is inefficient. The tracer tests revealed that minor amounts of water leave the catchment via slow groundwater pathways.

Stucki (1995) attempted to establish the provenance of the water that flows in the intra-permafrost bedrock talik at $52-55$ m (Sect. 2.1) using stable isotopes as tracers of possible water sources: glacier ice from nearby *Vadret da Corvatsch*, permafrost ice from the same 2/1987 drillcore sampled near the ground-ice table, precipitation, and nine springs in the catchment (among them $S_{EE}$, $S_E$, $S_W$, and $S_{WW}$). The $\delta^{18}$O of the ground ice was with $-15.3‰$ identical to that of the snow sample, the $\delta^{18}$O of the outflow $S_E$ became seasonally depleted from $-11‰$ (July 30, 1994) to $-14‰$ (Oct 7) at overall decreasing discharge, and their $\delta^{18}$O–$\delta^2$H relationship is $\delta^2$H $= 8.4\,\delta^{18}$O $+ 18.7‰$ ($n = 30$, $R^2 = 0.99$).

## 3  Measurements and data processing

A comprehensive hydro-meteorological data set is obtained from hydrological sensors (Table 1, Sect. 3.1), from the PERMA-XT and PERMOS automatic weather stations (AWS) on the rock glacier (Amschwand et al., 2024a) (Sect. 3.2), from active-layer sensors complemented with PERMOS borehole temperature data to estimate the AL energy budget (Amschwand et al., 2024b) (Sect. 3.3), and direct observation of the ground ice table (Sect. 3.4). Throughout this publication, the term 'meltwater' denotes melt from various types of ground ice, but neither from snow (explicitly denoted as 'snowmelt'), nor from glacier ice (the catchment is not glacierized).

### 3.1  Hydrological measurements

#### 3.1.1  Discharge measurement

The water level gauge is located in the lower forefield after the confluence of all four rock-glacier springs and captured the catchment-integrated surface outflow (Fig. 2, Table 1). The discharge $Q_w$ is expressed as a power-law relation of water level $h_w$ (stage) with empirically fitted coefficients $c_1$ and $c_2$,

$$Q_w = c_1(h_w - h_0)^{c_2}, \tag{1}$$

where $h_0$ is the stage at zero discharge (standing water in logger pool). The coefficients for the stage–discharge ($h_w$–$Q_w$) relation (Eq. 1) were constrained by dilution gaugings with sodium chloride as a chemical tracer or the volumetric method ("bucket method") if discharge was too low for dilution gaugings. The water level is obtained from the total pressure $P_{tot}$ measured by the submerged logger (pressure compensation) via the *hypsometric equation* that corrects the barometric pressure measured on the rock glacier to the elevation of the gauging station,

$$h_w = \frac{P_{tot} - P'_{atm}}{\rho_w\, g}, \quad P'_{atm} = P_{atm} \cdot \exp\left\{\frac{\Delta z\, g}{R \bar{T}_v}\right\}. \tag{2}$$

$\Delta z = 42$ m is the elevation difference, $\bar{T}_v$ [K] the layer-averaged virtual temperature (approximated by the actual temperature $T_a$), $P'_{atm}$ the elevation-corrected air pressure, $\rho_w = 10^3$ kg m$^{-3}$ the water density, $g$ the gravitational acceleration [9.81 m s$^{-2}$], and $R$ the specific gas constant [287 J kg$^{-1}$ K$^{-1}$].

**Table 1.** PERMA-XT sensor specifications. The below-ground sensors were operational from Sep 2020 to Sep 2023 (destroyed by rockfall).

| Quantity [unit] | Manufacturer | Sensor type | Accuracy |
|---|---|---|---|
| *Sensors above ground* (details in Amschwand et al. (2024a)) | | | |
| Air temperature $T_a$ [°C] | CSI[a] | 107 temperature probe[b] | $\pm 0.01$ °C |
| Relative humidity (rH for $q_a$) [%] | CSI | HygroVUE10 hygrometer[b] | $\pm 3\%$; $\pm 0.1$°C |
| Barometric pressure $P$ [Pa] | CSI/SETRA | CS100 barometer | $\pm 1.5$ hPa |
| Liquid precipitation $r$ [mm h$^{-1}$] | CSI | SBS500 tipping bucket rain gauge (unheated) | $\pm 30\%$ (undercatch) |
| Snow temperature [°C] | TE Connectivity[d] | 44031RC NTC thermistors (0, 25, 50, 100 cm a.g.l., unshielded) | $\pm 0.1$°C |
| Snow height $h_S$ [cm] | CSI | SR50A sonic ranging sensor | $\max\{\pm 1\,\text{cm}, \pm 0.4\%\}$ |
| Automatic time-lapse camera | MOBOTIX | M16B IP camera (RGB) | |
| *Sensors below ground* (active-layer sensors in instrumented main cavity, details in Amschwand et al. (2024b)) | | | |
| AL air temperature $T_{al}(z)$ [°C] | TE Connectivity[a] | 44031RC NTC thermistor chain TK1/1 | $\pm 0.1$°C |
| AL net long-wave radiation $Q_{\mathrm{CGR3}}^{rad}$ [W m$^{-2}$] | Kipp & Zonen | CGR3 pyrgeometer (4.5$-$42 µm, FoV 150°) | $< 4$ W m$^{-2}$ |
| Heat flux $Q_{\mathrm{HFP}}$ [W m$^{-2}$] | Hukseflux | HFP01 heat flux plate | site-specific |
| *Hydrological sensors* (on Murtèl forefield) | | | |
| Water level[c] (pressure $P_{tot}$) [Pa] | Onset | HOBO U20-001-04 water level logger | $\pm 0.43$ kPa ($\pm 3$ mm) |
| Water electrical conductivity[c] $\varkappa$ [µS cm$^{-1}$] | Onset | HOBO U24-001 conductivity logger | $\max\{3\%, \pm 5\,\text{µS cm}^{-1}\}$ |
| | Driesen+Kern | D+K µS-Log3040 | 2% FS |

Measurement range and accuracy by manufacturer/vendor. The specifications of the PERMOS sensor are given in Scherler et al. (2014) and Hoelzle et al. (2022).

[a] Thermistor strings manufactured by Waljag GmbH. [b] CSI: Campbell Scientific, Inc. Sampling interval: 30 minutes (or shorter). [c] All water sensors additionally measure temperature.

### 3.1.2 Water electrical conductivity monitoring

The electrical conductivity (EC) of the water is monitored at the two main springs at the rock-glacier front and of the total outflow in the bedrock step downstream of the confluence (Fig. 2). We report water EC referenced to 25°C, $\varkappa$ [µS cm$^{-1}$], calculated from the measured conductivity $\varkappa_\vartheta$ [µS cm$^{-1}$] at water temperature $\vartheta_w$ [°C] via

$$\varkappa = \frac{\varkappa_\vartheta}{1 + \alpha(\vartheta_w - 25°\mathrm{C})}, \tag{3}$$

with a linear temperature compensation factor $\alpha = 0.019$°C$^{-1}$ (Hayashi, 2004; McCleskey et al., 2011). Additionally, we measured EC manually using a WTW LF 320 with a TetraCon 325 probe as a reference for the two quasi-continuously measuring conductivity logger models (Table 1) and when the water loggers were found dry (water level too low).

### 3.1.3 Stable isotope composition

We took grab samples of the spring snowpack before onset of snowmelt (coring with a plastic tube), of the rock-glacier outflow, cumulative rainwater, and supra-permafrost water in a rock glacier furrow where the ground ice is accessible (next to the 'ablation stake'; Sect. 3.4, Fig 2). The water samples were stored in PP bottles with little head space and tightly sealed parafilm in order to minimize evaporation effects.

Water stable isotope composition ($\delta^2$H, $\delta^{18}$O) was analysed by Cavity Ring Down Spectroscopy at the Institute of Geology of the University of Bern using a Picarro L2120-i analyzer attached to a V1102-i vaporizer.

We report the water stable isotope composition as a $\delta$ ratio [‰] of the sample to the Vienna Standard Mean Ocean Water (VSMOW), where $\delta$ is the ratio of $^{18}$O/$^{16}$O and $^2$H/$^1$H. Analytical precision is $\pm 1.0$‰ for $\delta^2$H and $\pm 0.1$‰ for $\delta^{18}$O. The altitudinal gradient in $\delta^{18}O$ does not exceed $-0.2$‰ per 100 m (annual average) (IAEA/WMO, 2015; Bowen, 2017; Kern et al., 2014), i.e. the isotopic differences over the catchment elevation range (2600–3100 m asl.) is within 1‰.

## 3.2 Surface fluxes: Precipitation and evaporation

### 3.2.1 Snow

The PERMA-XT point-wise snow depth measurements $h_S$ (sonic ranger data, Table 1) located on a wind-swept rock-glacier ridge are converted to snow water equivalent (SWE) [kg m$^{-2}$ = mm w.e.] with the semi-empirical parsimonious $\Delta$SNOW model (Winkler et al., 2021b). Additionally, to harness the larger footprint of the SEB for a spatially averaged snow depth estimate on the rugged rock-glacier surface, the deviation of the SEB $\mathrm{dev}_{\mathrm{SEB}}$ during the snow melt months is back-calculated to SWE,

$$\mathrm{SWE} \approx \mathrm{dev}_{\mathrm{SEB}} \Delta t / L_m, \tag{4}$$

where $\mathrm{dev}_{\mathrm{SEB}} := Q^* - Q_H - Q_{LE} - Q_G$ (Amschwand et al., 2024a): The SEB deviation is the remainder of the turbulent fluxes $Q_H + Q_{LE}$ and ground heat flux $Q_G$ subtracted from the surface net radiation $Q^*$. $\Delta t$ refers here to the duration of the snowmelt period as estimated from temperature measurements in the snowpack ($0°$C) and using time-lapse imagery from an on-site camera (Table 1).

### 3.2.2 Rain

Liquid precipitation data is taken from the on-site rain gauge, assuming that precipitation is liquid based on a threshold air temperature of $T_{wb} = 2°$C. The rainfall heat flux $Q_{Pr}$ was estimated via (Sakai et al., 2004; Reid and Brock, 2010)

$$Q_{Pr} = C_w r (T_P - 0°C), \tag{5}$$

where $C_w = \rho_w c_w$ [4.18 MJ m$^{-3}$ K$^{-1}$] is the water volumetric heat capacity and $r$ [m$^3$ m$^{-2}$ s$^{-1}$] is the rainfall rate intercepted at the surface as measured by our on-site rain gauge or from MeteoSuisse data (Sect. 4.3). Precipitation temperature

$T_P$ was approximated using the wet-bulb temperature $T_{wb}$, calculated from air temperature and relative humidity (Amschwand et al., 2024a).

### 3.2.3 Evaporation/sublimation

The evaporative water flux (including sublimation if $T_s < 0\,°C$) is derived from the latent turbulent flux $Q_{LE}$ of the Amschwand et al. (2024a) SEB that is estimated with the the bulk aerodynamic method (Mittaz et al., 2000; Hoelzle et al., 2022). The flux–gradient relation is expressed as

$$Q_{LE} = \rho_a L_v \frac{q_a - q_s}{r_q}. \tag{6}$$

$Q_{LE}$ is driven by the specific humidity difference between the atmospheric air $q_a$ and the snow (if snow-covered) or debris surface (if snow free) $q_s$. $q_s$ under snow-free conditions is taken from humidity measurements in the near-surface AL ($q_a$ at $0.7\,m$ depth; Table 1), otherwise the surface is considered saturated at the radiometrically determined surface temperature. The bulk aerodynamic resistance for vapour transport in the near-surface atmosphere $r_q$ [s m$^{-1}$] decreases with the strength of turbulence: a thermally unstable atmosphere or strong winds enhance turbulent transport. $r_q$ is estimated using the Monin–Obukhov similarity theory (MOST) and the parameterisations detailed in Rigon et al. (2006); Endrizzi et al. (2014).

## 3.3 AL heat fluxes

### 3.3.1 The AL energy budget

During the thaw season, the ground heat flux $Q_g$ [W m$^{-2}$] downwards into the coarse-blocky AL is spent on warming the debris $\Delta H_{al}^\theta$ (sensible heat storage changes), melting ground ice in the AL $\Delta H_{al}^i$ (latent heat storage change or ground ice storage change), and conducted into the permafrost body beneath $Q_{PF}$ ('permafrost heat flux') (cf. Woo and Xia, 1996; Hayashi et al., 2007; Boike et al., 2003; Zhu et al., 2024) (Fig. 3),

$$\underbrace{\frac{\partial}{\partial t} \int_0^{\zeta(t)} C_v \left(T_{al}(z,t) - 0°C\right) dz}_{\Delta H_{al}^\theta} + \underbrace{L_m \rho_i \frac{\partial}{\partial t} \int_0^{\zeta(t)} f_i(z) dz}_{\Delta H_{al}^i,\ \mathrm{dev}_{al},\ Q_m} = \underbrace{Q_g - Q_{PF}}_{Q^{\mathrm{net}}} \ [\mathrm{W\ m}^{-2}]. \tag{7}$$

where $\zeta$ is the depth of the ground-ice table (AL thickness) [m], $C_v$ the volumetric heat capacity of of the debris [J m$^{-3}$ K$^{-1}$], and $T_{al}$ AL temperatures [°C]. $f_i$ [−], $L_m$ [$3.35 \times 10^5$ J kg$^{-1}$], and $\rho_i$ [kg m$^{-3}$] are the volumetric ice content, latent heat of melting, and ice density, respectively. The AL energy budget Eq. 7, derived from the conservation of energy principle, is estimated based on data from the instrumented main cavity (Fig. 2) as outlined below. We compare two independent estimates of the ground ice melt $\Delta H_{al}^i$, one from the deviation of the AL energy budget denoted by $\mathrm{dev}_{al}$ (Sect. 3.3.3), and one from the Stefan model based on direct observations of $\zeta(t)$ in a nearby rock-glacier furrow ('ablation measurements') denoted by $Q_m$ (Fig. 2, Sect. 3.4). Details on the measurement set-up and data processing are in Amschwand et al. (2024a, b).

### 3.3.2 Ground heat flux $Q_g$

We estimate the thaw-season ground heat flux from two measurements, from the AL net long-wave radiation $Q_{\mathrm{CGR3}}^{rad}$ and the heat flux plate $Q_{\mathrm{HFP}}$ (Table 1). These two measurements are correlated and represent the downward heat flux $Q_g$ in the absence of buoyancy-driven convection, i.e. in conditions of stably stratified AL air column which prevails during the thaw season. These $Q_g$ measurements are at $1.5-2.0$ m depth in the AL, not at the surface. Details about data pre-processing are in Amschwand et al. (2024b).

### 3.3.3 Sensible and latent heat storage changes $\Delta H_{al}^{\theta}$, $\mathrm{dev}_{al}$

The sensible heat $\Delta H_{al}^{\theta}$ stored/released by temperature changes of the blocks in the coarse-blocky AL beneath the $Q_g$ measurement depth are estimated by

$$\Delta H_{al}^{\theta} \approx C_v h \frac{\partial \langle T_{al} - 0°\mathrm{C}\rangle}{\partial t} \tag{8}$$

where $C_v = (1 - \phi_{al})\rho_r c_r$ is the volumetric heat capacity [$0.6 \times 2690$ kg m$^{-3} \times 780$ J kg$^{-1}$ K$^{-1}$] (porosity $\phi_{al} = 0.4$), $h$ the distance from the $Q_g$–measurement level to the AL base [2 m], and $\langle T_{al}\rangle$ spatially averaged AL temperatures [°C].

The deviation $\mathrm{dev}_{al}$ to closure of the AL energy budget (Eq. 7), after assessment of the uncertainties, corresponds to the latent heat storage changes, i.e. heat spent on melting ground ice,

$$\mathrm{dev}_{al} := (Q_g - Q_{\mathrm{PF}}) - \Delta H_{al}^{\theta}. \tag{9}$$

### 3.3.4 AL base flux through permafrost body $Q_{\mathrm{PF}}$

The heat flux across the permafrost table $Q_{\mathrm{PF}}$ is estimated with the gradient method from PERMOS borehole temperature data via Fourier's heat conduction equation

$$Q_{\mathrm{PF}} \approx -k_{\mathrm{PF}} \frac{\Delta T_{\mathrm{PF}}}{\Delta z}, \tag{10}$$

where the borehole temperatures are measured at 4 and 5 m depth in the permafrost body beneath the AL. We take a thermal conductivity $k_{\mathrm{PF}}$ value of $2.5$ W m$^{-1}$ K$^{-1}$ (Vonder Mühll and Haeberli, 1990; Scherler et al., 2014).

## 3.4 'Ablation measurements' at ground-ice table

### 3.4.1 Observations of seasonal evolution of the ground-ice table

The ground ice is accessible at a few spots, all located in furrows where the AL is thinner ($1-2$ m). In one spot, a plastic tube was drilled ca. $120$ cm into the ice in August 2009 but subsequently abandoned (C. Hilbich, pers. comm.). We made serendipitous use of it as an 'ablation stake', manually measuring the depth of the ground-ice table $\zeta(t)$ [m] at each field visit in summer 2022 and 2023 (Amschwand et al., 2024b). Assuming that changes in ice content $f_i$ only occur at the phase change

boundary $\zeta(t)$ coinciding with the $0°C$ isotherm (negligible melting point depression in the coarse material), the ice melt heat flux $Q_m$ can be expressed as

$$Q_m = f_i L_m \rho_i \frac{d\zeta}{dt}. \tag{11}$$

### 3.4.2 Stefan parameterisation of ground-ice melt

Given the sparse and point-wise ablation observations, we use a Monte Carlo simulation and a Stefan model (Eq. 12) to assess a plausible range of ground-ice melt for a range of input parameter values as expected on landform scale (probabilistic uncertainty estimate, Sect. 4.4.2). The Stefan model has been widely used to simulate the freezing and thawing fronts in permafrost (e.g., Hayashi et al., 2007; Riseborough et al., 2008; Bonnaventure and Lamoureux, 2013; Hrbáček and Uxa, 2019), including the Cold Regions Hydrological Model (Pomeroy et al., 2022). We parameterise the cumulative heat flux from ground ice melt on Murtèl $H_{al}^i(t) = \int_0^t Q_m dt' = f_i L_m \rho_i \zeta(t)$ as a function of the depth of the ground ice table $\zeta(t)$ using a modified Stefan equation by Aldrich and Paynter (1953) that is appropriate for a two-layered stratigraphy (Fig. 3) (Kurylyk, 2015; Kurylyk and Hayashi, 2016),

$$\int_0^t Q_m(t') dt' := \Sigma_t Q_m = f_i L_m \rho_i \sqrt{h_1^2 + \frac{2k_{\text{eff}}(I(t) - I_1)}{L_m f_2 \rho_i}}, \tag{12}$$

where the surface thawing index $I(t)$ is defined as

$$I(t) := \int_0^t \lambda_5^2 T_s dt' \approx 86400 \sum_i \bar{\lambda}_5^2 \bar{T}_s \tag{13}$$

and the thaw index of the ice-poor AL overburden $I_1$ as

$$I_1 := \frac{h_1^2 L_m f_1 \rho_i}{2k_{\text{eff}}}. \tag{14}$$

The factor $\lambda_5 \leq 1$ corrects for the sensible heat storage in the thawed layer and is a polynomial of the Stefan number Ste, $\lambda_5 = 1 - 0.16 \,\text{Ste} + 0.038 \,\text{Ste}^2$ (Kurylyk and Hayashi, 2016). The depth-averaged dimensionless Stefan number Ste is proportional to the ratio of sensible heat to latent heat absorbed during thawing,

$$\text{Ste} := \frac{C_v \bar{T}_s}{L_m \langle f \rangle \rho_i}, \tag{15}$$

with the bulk volumetric heat capacity $C_v$ [J m$^{-3}$ °C$^{-1}$] of the (unfrozen, ice-free) AL (identical for both layers), the average surface temperature $\bar{T}_s$ for the time $t$ elapsed since onset of the thaw season, and the latent heat consumed by the melting of the ground ice $L_m \langle f \rangle \rho_i$ (different in each layer and depth-averaged denoted by $\langle \cdot \rangle$; details in Kurylyk and Hayashi (2016)).

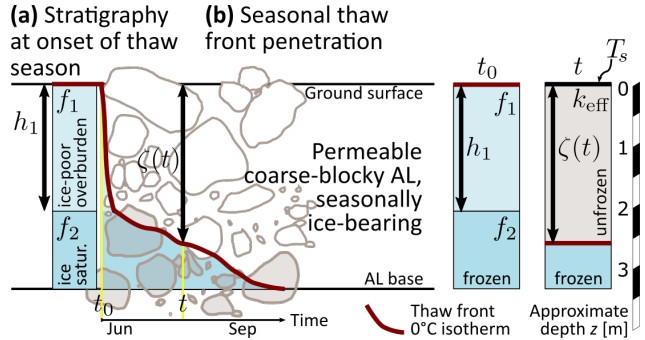

**Figure 3.** Ground-ice thaw and the Stefan equation of a two-layered stratigraphy. **(a)** Initial stratigraphy at the onset of the thaw season with ice-poor overburden (thickness $h_1$ and ice content $f_1$) and ice-saturated layer (ice content $f_2 = \phi_{al}$). **(b)** Seasonal thaw front penetration, $\zeta(t)$, driven by the ground surface temperature $T_s$. Maximum thaw depth is denoted by $\zeta_{max}$.

## 4  Results

### 4.1  Meteorological conditions

The weather in each season differed markedly in the two years 2020–2022. Figs. 4a and 5a show temperature and precipitation during the summers 2021 and 2022, respectively. The winter 2020–2021 was colder than the 2021–2022 one (November–April: average temperature: $-6.2°$C vs. $-5.3°$C; minimum daily average temperature: $-16.5°$C vs. $-15.1°$C) and richer in terms of snow amount (November–April: average snow height measured on a wind-swept ridge: $76\,\mathrm{cm}$ vs. $54\,\mathrm{cm}$) and duration (early onset of snow cover: 5 October vs. 3 November; later melt-out: mid-June vs. mid-May). Summer 2021 was cool-wet compared with the hot-dry summer 2022; temperatures were lower (July–August: average: $6.9°$C vs. $9.3°$C) with frequent passage of synoptic fronts, often bringing cold air ($\leq 3°$C; minimum daily average temperature: $0.7°$C vs. $5.6°$C) and mixed precipitation (sleet). Snowfall occurred in a few days throughout the summer and melted within hours. A few snow patches survived over the summer after melt-out of the winter snowpack in mid-June (Fig. 2), which has rarely been occurring in the last $\sim 15$ years. A thermistor installed above the ground-ice table in August 2020 became embedded into newly formed ground ice and was only released in August 2022. In contrast, the hot-dry summer 2022 was marked by three heat waves (in June, July and August) and daily minimum temperatures not below $5°$C. Several dry spells occurred during this season; the longest one was an 11-day long dry spell within the 5–19 July heat wave. Almost no precipitation was recorded between 20 June and 1 August, despite strong convective precipitation events recorded on by the nearby MeteoSuisse station *Piz Corvatsch* (3294.31 m asl., 1.2 km away). Discharge data of the rock glacier outflow (Fig. 5b), camera images and post-event field observations (fresh debris flow deposits, flooding of furrows) revealed rainwater funnelled onto the rock glacier. We augment the PERMA-XT precipitation measurements with MeteoSuisse precipitation data from the station *Piz Corvatsch*. Immediate on-site inspection of the PERMA-XT rain gauge did not suggest any technical malfunctioning, speaking for a spatially heterogeneous precipitation pattern (Sect. 4.3).

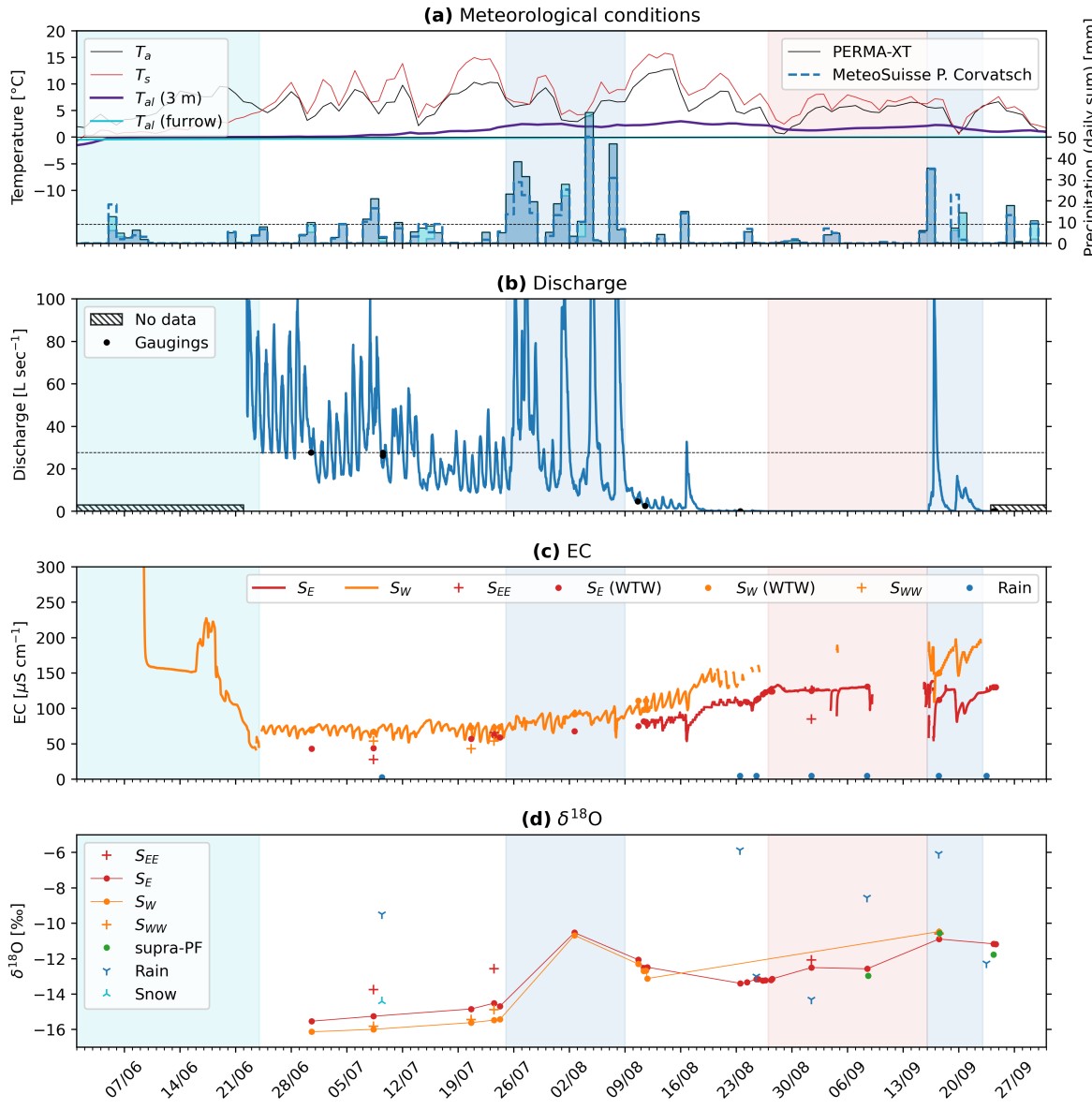

**Figure 4.** Hydro-meteorologial data for summer 2021. **(a)** Temperature and precipitation (daily averages). Thermistor TK4/5 $T_{al} = 0°C$ in the furrow remained frozen. $9\,\text{mm w.e. day}^{-1}$ is the infiltration capacity (horizontal dashed line, Fig. 7). **(b)** Discharge. Maximum gauged discharge is $27.7\,\text{L s}^{-1}$ (horizontal dashed line). **(c)** Water electrical conductivity (EC). **(d)** $\delta^{18}O$.

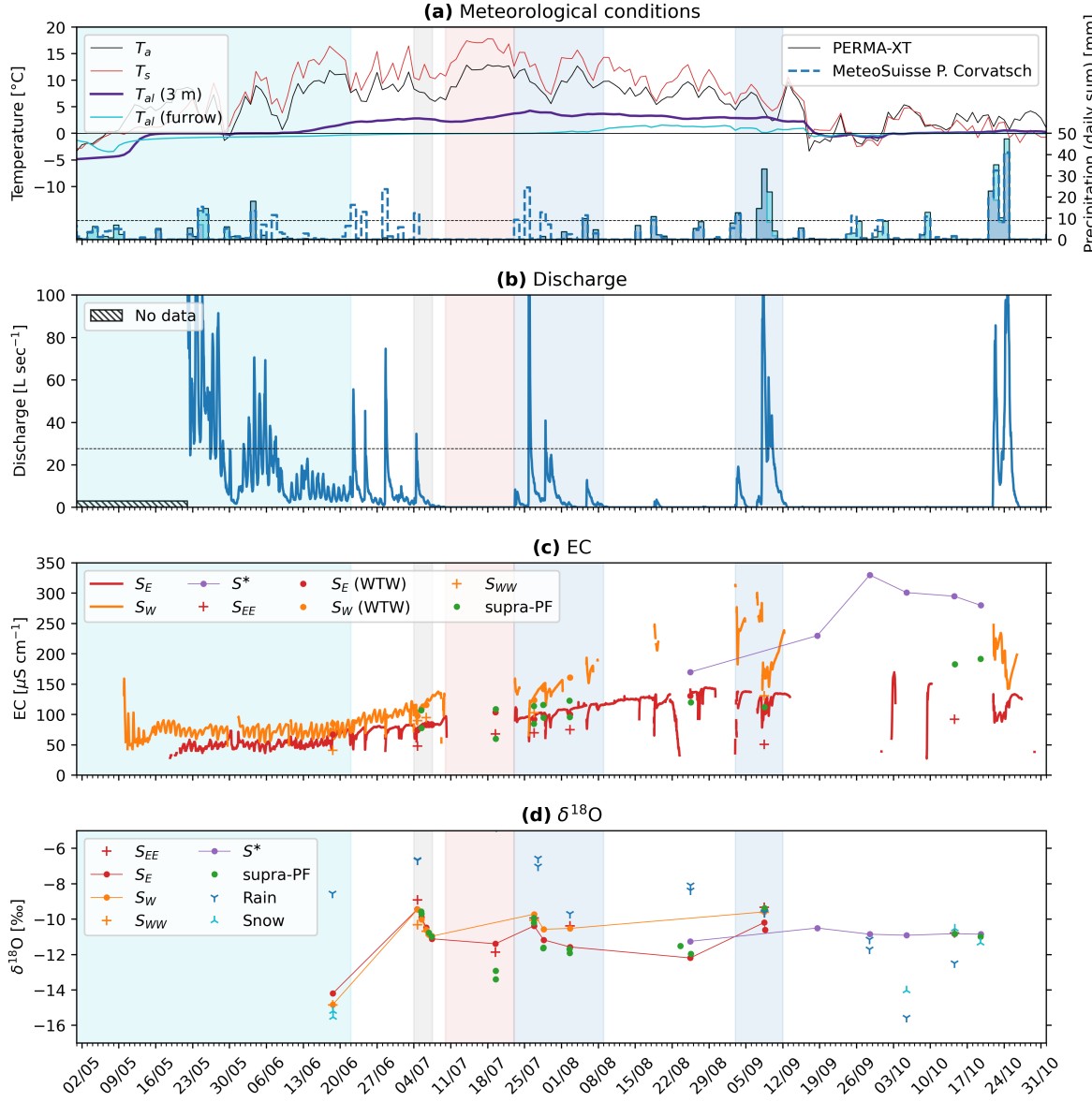

**Figure 5.** Hydro-meterologial data for summer 2022.

## 4.2 Hydrological results

Frequent field visits were indispensable to obtain the in-situ measurements, to obtain grab samples of water between the coarse-blocky material, and to adapt the logger placement to the strongly variable discharge. Suitable places for the EC loggers at the two main springs were found only in late summer 2021. Still, data gaps due to incompletely submerged loggers at extremely

low outflow during hot–dry periods (precisely when meltwater signal can be expected to be clearest) could not be avoided. Six snapshots of the strongly variable discharge in the Murtèl rock glacier forefield are drawn in Fig. A2. All springs are mentioned in previous investigations (Tenthorey and Gerber, 1991; Stucki, 1995).

### 4.2.1 Discharge and water temperature

The empirical stage–discharge ($h_w$–$Q$) relation (Eq. 1) based on eight gaugings (Fig. 6, Table 2) yields the fitted coefficients $h_0 = 780 \, \mathrm{mm}$ (stage of the standing water), $c_1 = 7429$, and $c_2 = 5.15 \pm 0.753$ ($1\sigma$ uncertainties; $h_w$ in m, $Q_w$ in $\mathrm{m^3 \, s^{-1}}$). Because of the wide channel (plane-bed type stream morphology) in the slightly dipping forefield, the water level covered by gaugings varies by merely $9 \, \mathrm{cm}$ that covers a discharge range from $3 \, \mathrm{L \, min^{-1}}$ (detection threshold) to $27.7 \, \mathrm{L \, sec^{-1}}$ (Fig. 6). The channel remained stable during the study period August 2020–September 2022. Discharge estimates in the wet summer

2021 relies on extrapolated stage–discharge relation (often exceeding $27.7 \, \mathrm{L \, sec^{-1}}$), while discharge in the dry summer 2022 is mostly interpolated and better constrained (except early snow melt period and peak discharge of event water). Importantly, no stage measurements could be made beneath a snow cover, hence snowmelt discharge before complete melt-out of the forefield is not gauged. We consider our stage–discharge relation and discharge estimate sufficient for our purpose of season-cumulative water balances and emphasize that our focus is on the low-discharge summer periods where the contribution from ground-

ice melt is potentially largest. In the context of the hydrological significance of Murtèl rock glacier, the (reliably measured) zero-discharge estimates will be important.

Discharge in the small catchment is variable and shows a seasonal trend and decreases as snowmelt progresses (Figs. 4, 5), superimposed by regular diurnal fluctuations related to radiative forcing/snowmelt. Total measured discharge was $160 \times 10^3 \, \mathrm{m^3}$ in summer 2021 (snowmelt and thaw season), and $97 \times 10^3 \, \mathrm{m^3}$ in summer 2022. After completion of the snowmelt, outflow

is 'flashy' where dry phases without measurable baseflow ($\lesssim 3 \, \mathrm{L \, min^{-1}}$) are interrupted by precipitation-fed discharge spikes (event water). The qualitative discharge pattern is similar in both summers. Field observations and additional EC measurements at the bedrock step downstream of the confluence (mixing calculations) suggest that the discharge of $S_W$ exceeds that of $S_E$, if $S_W$ is active. Water temperature of the outflow and the deep seep are stable and always at $0-1°\mathrm{C}$.

The rainfall–streamflow relation from the *threshold analysis* (Wagner et al., 2020; Harrington et al., 2018) (Fig. 7) is cal-

culated for the late-summer discharge after completion of the snowmelt in the entire catchment. For our purpose, only one observation is relevant: Rainfall less than $9 \, \mathrm{mm \, day^{-1}}$ in most cases (8/10) does not generate measurable outflow at the gauging station (and this finding is independent of the quality of the stage–discharge relation). This agrees with field observations: Total spring discharge below the detection threshold of $\sim 3 \, \mathrm{L \, min^{-1}}$ seeps into the ground along its way from the rock-glacier springs to the gauging station located $\sim 50 \, \mathrm{m}$ below the rock-glacier front ("not measurable baseflow" refers to maximal

discharge of $\sim 3 \, \mathrm{L \, min^{-1}}$). We observed rapid recession and drying out of the stilling pool at the gauging station after the discharge estimate in the morning of Aug 24, 2021 (Fig. A2c, Table 2). We so constrained the detection limit of $\sim 3 \, \mathrm{L \, min^{-1}}$ using the bucket method (discharge too small for dilution gaugings).

**Table 2.** Discharge measurements and observations in summer 2021.

| Date (CEST time) | Stage[a] [mm] | Discharge[b] [L s$^{-1}$] | Method |
|---|---|---|---|
| Jun 30 (15:00–15:42) | 89 | 27.7 ± 1.3 | dilution gauging |
| Jul 9 (08:45–09:41) | 88 | 27.6±1.6 | dilution gauging |
| Jul 9 (09:46–10:33) | 85 | 26.1±1.9 | dilution gauging |
| Aug 10 (14:07–16:14) | 67 | 4.7±0.05 | dilution gauging |
| Aug 11 (10:37–13:45) | 56 | 2.6±0.3 | dilution gauging |
| Aug 23 (14:30) | 22 | 3.6±0.4 L min$^{-1}$ | volumetric method |
| Aug 24 (09:00) | 17 | 3.1±0.3 L min$^{-1}$ | volumetric method |
| Aug 24 (15:00)[c] | 0.0 | $\lessapprox$ 3 L min$^{-1}$ | field observation |

[a] Stage relative to stage $h_0$. Analytical stage uncertainty of stage measurement: ±3 mm (Table 1). [b] Analytical discharge uncertainty: standard deviation from three simultaneous EC measurements; 10% of bucket measurement. [c] Spring discharge of $\lessapprox$ 3 L min$^{-1}$ seeps away between the spring and the gauging station (detection threshold), the logger pond falls dry (Fig. A2c).

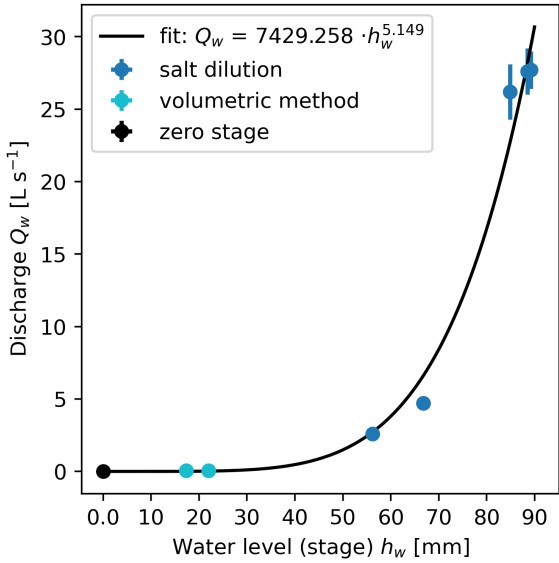

**Figure 6.** Stage-discharge relation as established with dilution gaugings and bucket measurements in summer 2021 (Table 2).

### 4.2.2 Water electrical conductivity

EC of the outflow seasonally increases from $\sim$ 50 µS cm$^{-1}$ to $\sim$ 150 µS cm$^{-1}$ (Figs. 4, 5). Both main springs $S_E$ and $S_W$
show a qualitatively similar behaviour in both summers, despite different weather conditions: They transition from a snowmelt-regime (daily oscillations) to a rain-regime (rapid EC drop after onset of event discharge that stabilises). The lower-lying main

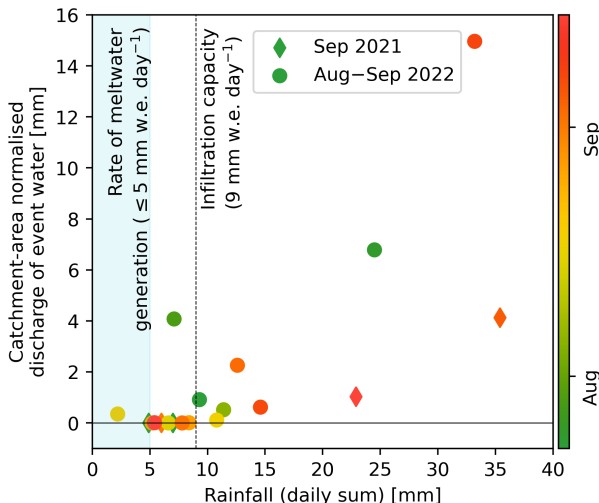

**Figure 7.** Rainfall–streamflow relation (threshold analysis). Rainfall less than $9\,\mathrm{mm\,day^{-1}}$ does not trigger surface outflow (infiltration capacity).

spring $S_E$ (2624 m asl.) has a lower EC than $S_W$ (2626 m asl.) and appears more strongly buffered in terms of smaller daily oscillations and smaller seasonal shift. Still, the different weather conditions might show up in the late-season EC: The $S_E$–$S_W$ EC difference is largest in autumn 2022 after the dry hydrological year 2021–2022 (SWE and summer rainfalls below average;

Fig. 9), where the EC of $S_W$ reaches $> 250\,\mathrm{\mu S\,cm^{-1}}$. EC is in general anti-correlated with discharge from seasonal down to hourly timescales suggesting dilution behaviour or longer water residence times: High EC at low discharge which is best seen during snowmelt, although the hourly EC evolution during late-summer rainfalls can be complex with a high-EC peak at the onset of event-water discharge (a timescale beyond the scope of this study). Point-measurements of the seep $S^*$ show a strong enrichment up to $250-350\,\mathrm{\mu S\,cm^{-1}}$ of the autumn seep water. The supra-permafrost water co-evolves with the outflow.

Notable are the persistent EC difference between two nearby sampling spots in the same rock-glacier furrow, one next to the 'ablation stake', the other next to thermistor TK4/5 (Fig. 2). Spatially varying EC speaks for channelised supra-permafrost flow as interpreted from tracer tests by Tenthorey and Gerber (1991).

### 4.2.3 Stable isotope signature

We collected a total of 145 samples from different rock glacier springs and seeps (referred to as outflow), supra-permafrost

water, snowpack, and rainwater: 2 samples in 2020, 57 samples collected in 2021, and 86 in 2022 (Tables B1–B5). All but a few rainfall samples are aligned on our local meteoric water line (LMWL) given by $\delta^2\mathrm{H} = 8.19\,\delta^{18}\mathrm{O} + 15.35‰$ ($R^2 = 0.9919$) (Fig. 8). The $\delta^{18}\mathrm{O}$–$\delta^2\mathrm{H}$ relationship of the outflow samples is $\delta^2\mathrm{H} = 8.06\,\delta^{18}\mathrm{O} + 13.78‰$ ($R^2 = 0.9969$), with its slope of $8.1$ similar to the local (LMWL, $8.2$) and global meteoric water line (GMWL, $8.0$), suggests that the source waters of the outflow have undergone little if any evaporation. This finding is consistent with a sparsely vegetated coarse-blocky landform with rapid

infiltration (Williams et al., 2006; Krainer et al., 2007), and it is consistent with the measured specific humidity gradients in the Murtèl AL (Amschwand et al., 2024a, b): Moisture for evaporation is (generally) drawn from a rain-fed reservoir in the shallow AL, not from the supra-permafrost water in the deep AL. Moisture transport in the AL is generally downwards leading to condensation; upwards transport and evaporation from the deep AL occurs only episodically during droughts.

$\delta^{18}$O of the outflow and supra-permafrost water showed an enrichment during the thaw season from $-16‰$, also measured in the spring snowpack, to $-10‰$, levelling off in late summer–autumn, and repeatedly interrupted by short excursions towards isotopically 'heavier' values of $-10‰$ that co-occur with the isotopically enriched rainfall (typically $-10$ to $-6‰$). $\delta^{18}$O was overall higher in summer 2022 and reached the plateau phase sooner (by July), likely reflecting a proportionally smaller amount of snowmelt in the catchment after the snow-poor winter 2021–2022 (consistent with the EC pattern discused above). We do not observe the seasonal late-summer isotopic depletion reported by Stucki (1995). The two main springs $S_E$ and $S_W$ showed overall the same pattern, although $S_W$ showed seasonally somewhat more extreme values, i.e. isotopically 'lighter' than $S_E$ during snowmelt and 'heavier' in late summer. Discrepancies were smallest at high discharge during major rainfall periods. $\delta^{18}$O of the supra-permafrost water was always close to the outflow $\delta^{18}$O. Analogous to its EC, $\delta^{18}$O value of the supra-permafrost water in the eastern stretch of the furrow is closer to the eastern main spring $S_E$, while the one in the western stretch of the furrow (next to the 'ablation stake') is often closer to spring $S_W$ (if active). $\delta^{18}$O of the rainwater varied considerably between $-13‰$ and $-6‰$, but were generally 'heavier' than all other sampled waters. The denser 2022 data set shows the well-known seasonal pattern with maximum enrichment in July–August. The pattern agreed with 1994–2022 long-term measurements from the nearby Global Network for Isotopes in Precipitation (GNIP) station in Pontresina (1724 m asl.) with $\delta^2$H $= 8.095\,\delta^{18}$O$+9.53‰$ ($n = 321$, $R^2 = 0.9943$) (accessible via https://nucleus.iaea.org/wiser/ (IAEA/WMO, 2023)). $\delta^{18}$O of the snowpack were from $-15‰$ to $-19‰$. Although snow $\delta^{18}$O is sensitive to the sampling timing (Beria et al., 2018), its $\delta^{18}$O does not exceed outflow $\delta^{18}$O and is meaningful as a qualitative end member. The deuterium excess, in cases used as in indicator of multiple freeze-thaw cycles (Williams et al., 2006; Steig et al., 1998; Liaudat Trombotto et al., 2020; Munroe and Handwerger, 2023a, b), shows no clear seasonal trend (Fig. A4).

### 4.3 Precipitation (snow and rain) and evaporation/sublimation

The plot-scale water balance (Fig. 9) refers to the point-wise (not areal) ablation observations, precipitation measurements and measurements for the AL energy budget. For the precipitation, we compare different measurements to obtain a spatially representative value (Sect. 4.1):

- We augment the on-site PERMA-XT rainfall data with MeteoSuisse data from the nearby station *Piz Corvatsch*. Rainfall from these stations reasonably agree (Fig. A3) except during a "dry window" in July 2022 where no on-site precipitation was recorded, but rock glacier outflow occurred whose timing coincides with the MeteoSuisse measurements (Fig. 5a, b). Rain-on-snow events are not considered. The two thaw season differed in terms of cumulative precipitation: $460-500$ mm in the cool-moist 2021, and $\sim 320$ mm in the hot–dry 2022.

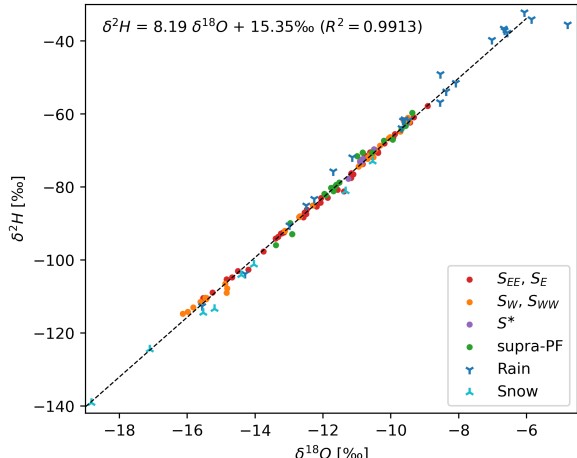

**Figure 8.** Dual isotope plot: $\delta^{18}$O–$\delta^2$H relationship of the outflow, supra-permafrost water, rainfall and snowpack.

– We use the SEB deviation during the snow melt period to obtain a representative SWE estimate ('SWE $\mathrm{dev_{SEB}}$' in Fig. 9). Comparison with the PERMOS snow height data and time-lapse images shows that the PERMA-XT snow depth measurement located on a wind-swept ridge grossly underestimates the average SWE on the rugged terrain (at least by a factor of $2.3$). Total SWE was $915\,\mathrm{mm}$ in winter 2020–2021 (average), but only $600\,\mathrm{mm}$ in the snow-poor winter 2021–2022.

The evaporative water flux (Fig. 9) during the thaw season is $\leq 3\,\mathrm{mm\,day^{-1}}$, amounting to $90\,\mathrm{mm}$ in the thaw season 2021 and $120\,\mathrm{mm}$ in 2022. Two remarks: First, we consider these values as upper bounds, since the SEB parameterisation in Amschwand et al. (2024a) likely tends to overestimate $Q_{LE}$. The additional aerodynamic resistance arising from the vapour transport within the coarse-blocky AL is unknown and ignored in Eq. 6, the more so, the deeper the moisture is drawn from, i.e. the most during dry spells. Second, thaw-season vapour transport in the coarse-blocky AL is generally downwards, the specific humidity gradient is aligned with the temperature gradient. It is not the supra-permafrost water that evaporates but meteoric water from the uppermost AL, except during dry spells when the rain-fed moisture store in the shallow AL is exhausted and the specific humidity gradient reverses (Amschwand et al., 2024a). In contrast to fine-grained material (or blocky material containing a fine-grained matrix), there is no upward transport of liquid water by capillary suction (Pérez, 1998).

## 4.4 Ground ice storage changes from in-situ measurements

### 4.4.1 Point-scale observations in the AL

The ground ice is rarely accessible in coarse-blocky landforms. Here, we present one of the worldwide few (to the best of our knowledge) direct observations of the seasonal evolution of superimposed AL ice in rock glaciers, ice that forms each spring from refreezing of percolating melt water. The ground ice table as observed in a rock glacier furrow in the thaw seasons 2022

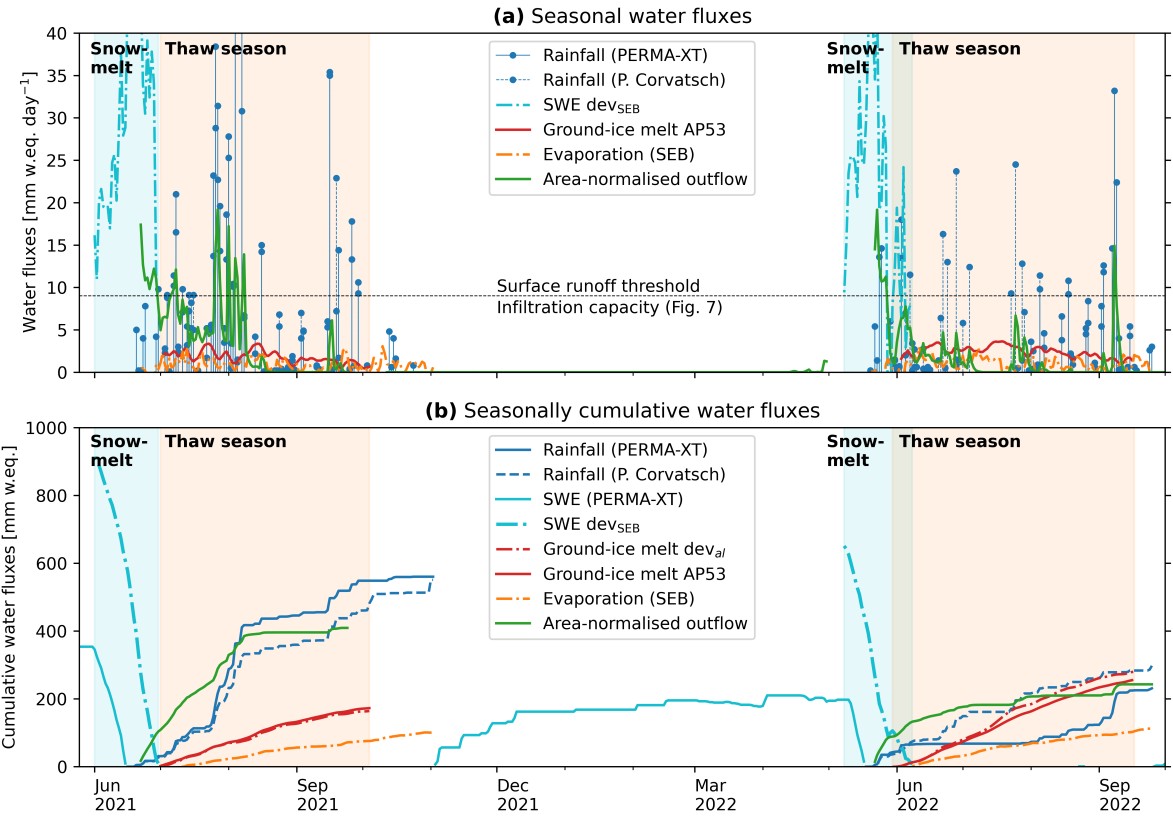

**Figure 9.** Murtèl water balance on a plot scale (daily average water fluxes in L m$^{-2}$ or mm w.e.). **(a)** Seasonal water fluxes. **(b)** Seasonally cumulative water fluxes.

and 2023 underwent seasonal accumulation and melt of $\sim 70\,\mathrm{cm}$ within the coarse-blocky AL, showing the seasonal build-up and melt of superimposed AL ice (Fig. 10). The seasonal release of water from melting ground ice in the coarse-blocky AL is $200-300\,\mathrm{L\,m^{-2}}$ over a thaw period of $\sim 100\,\mathrm{days}$, corresponding to $250\,\mathrm{mm}$ water equivalent (w.e.) or a melt rate of $1-4\,\mathrm{mm\ w.e.\ day^{-1}}$ $(1-4\,\mathrm{kg\,m^{-2}\,day^{-1}})$.

Our observations indicate two different mechanism of ground ice accumulation: (1) refreezing of snowmelt onto cold blocks (analogous to the formation of basal ice at the ground–snow interface), and (2) blowing in of snow with subsequent metamorphosis to ice. The 'warming spikes', the AL energy budget and ice coatings/icicles (Herz, 2006) in sheltered cavities indicate the former, while the observation of ripe snow on top of the fresh ground ice in exposed cavities indicate the latter mechanism.

### 4.4.2    Stefan model and probabilistic uncertainty estimate

The Stefan model (Eq. 12) describes the observed lowering of the ground-ice table ('Stefan AP53' in Fig. 10) and relates it to a modelled ground ice melt $Q_m$. The effective thermal conductivity $k_{\mathrm{eff}} = 3\,\mathrm{W\,m^{-1}\,K^{-1}}$ is derived from the heat flux

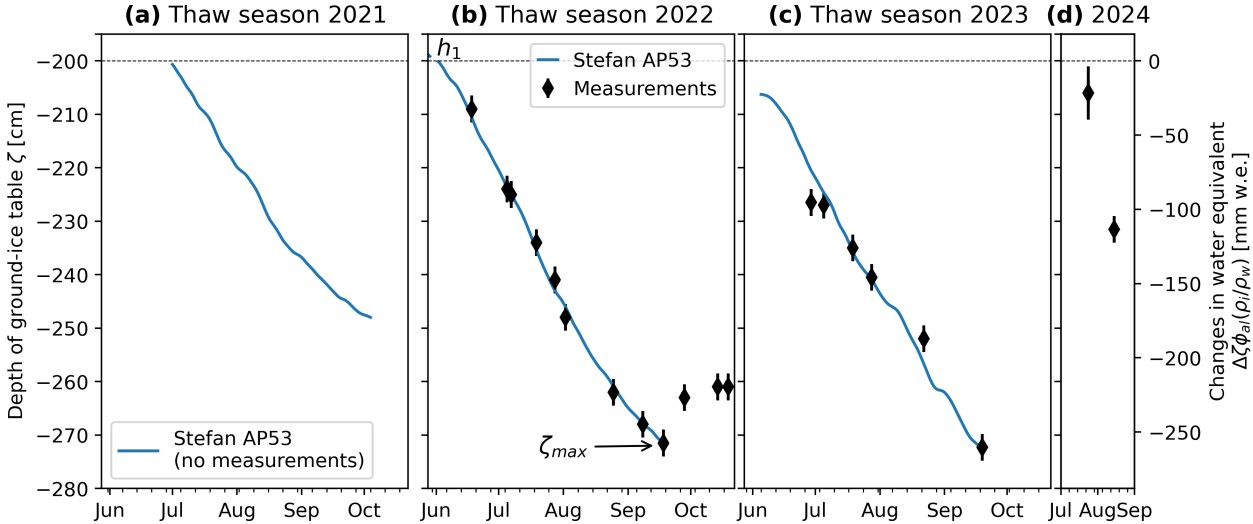

**Figure 10.** Observed and modelled vertical changes in the ground-ice table in thaw season **(a)** 2021 (modelled only), **(b)** 2022, **(c)** 2023, and **(d)** 2024 with seasonal accumulation and melt. Ice melt is simulated with the Stefan model (Eq. 12).

measurements (Amschwand et al., 2024b), the ice-poor AL overburden thickness $h_1 = 3$ m and ice content $f_1 = 0.01$ is calibrated with the 2022 ablation measurements. This 2022 parameter set also describes the 2023 ablation (Fig. 10c). The ablation observation-derived ground ice melt $Q_m$ is shown in Fig. 9 ("Ground-ice melt AP53") alongside the AL energy-budget estimate ($\text{dev}_{al}$, Sect. 4.5). Systematic ablation observations were not performed in summer 2021. The modelled 2021 ablation qualitatively agrees with observations insofar as a nearby thermistor (TK4/5) remained ice-embedded in summer 2021 and was only released in July 2022 ($T_{al}$ in furrow shown in Figs. 4, 5).

Given the sparse and point-wise observations on Murtèl and the few other published observations of seasonal ground ice melt in coarse-blocky landforms, we estimate the uncertainty with a probabilistic Monte Carlo approach. We make an educated guess of the value range of the input parameters the Stefan melt parameterisation is most sensitive to (Eq. 12), namely the effective thermal conductivity $k_{\text{eff}}$ ($2.0-3.5$ W m$^{-1}$ K$^{-1}$, Fig. 11c, Amschwand et al. (2024b)), the AL overburden thickness $h_1$ ($1.5-4.0$ m, Fig. 11d), and ice content $f_2$ ($0.2-0.8$, Fig. 11e), assuming that these parameters are independent from each other and to the ground surface temperature $T_s$. Fifty thousand model runs with the 2021 and 2022 $T_s$ forcing to represent two contrasting thaw seasons yield the outcome distribution of maximum thaw depth $\zeta_{max}$ and the total amount of ground ice melt (Fig. 11b, a). The expected amount of meltwater released in summer 2021 is $100-200$ mm w.e., and $150-350$ mm w.e. in summer 2022, with a large range of overlapping values of $150-250$ mm w.e.

### 4.5 Ground ice storage changes from the AL energy budget

The ground heat flux estimates were gained in the instrumented cavity (Fig. 2). Downward heat flux during the thaw season is typically $10-15$ W m$^{-2}$ as measured by the pyrgeometer $Q_{\text{CGR3}}$ and the heat flux plate $Q_{\text{HFP}}$ (Fig. 12a), amounting to

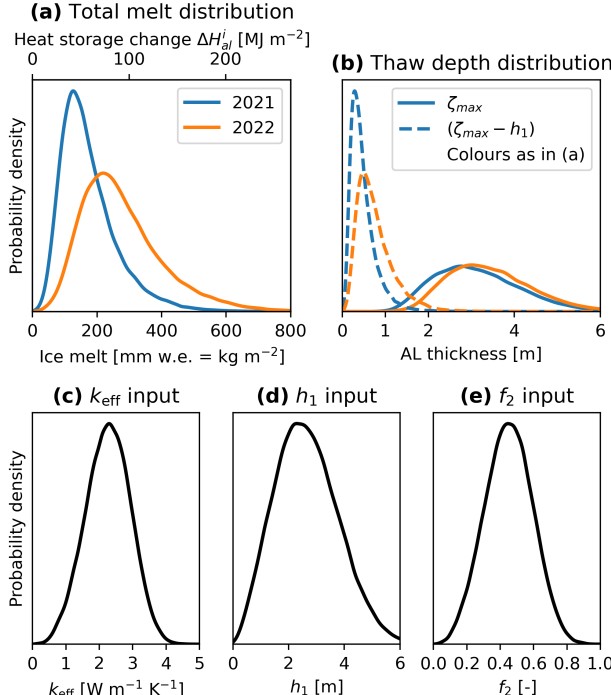

**Figure 11.** Probabilistic uncertainty estimate: Distribution of the thaw-season cumulative **(a)** ground-ice melt and **(b)** maximum thaw depth $\zeta_{max}$ calculated with the 2021 (wet-moist summer) and 2022 (hot–dry summer) $T_s$ forcing (Eq. 12) and a range of AL properties $(k_{\mathrm{eff}}, h_1, f_2)$ expressed by their input distributions **(c)** $k_{\mathrm{eff}}$, **(d)** $h_1$, and **(e)** $f_2$. Variables as in Fig. 3.

$40-60\,\mathrm{MJ\,m^{-2}}$ in cool–wet thaw season 2021, and $75-95\,\mathrm{MJ\,m^{-2}}$ in hot–dry thaw season 2022 (Fig. 12b, c). The rain heat flux $Q_{Pr}$ adds another $5-15\,\mathrm{MJ\,m^{-2}}$. The thaw-season cumulative heat uptake corresponds to less than $10\%$ of the net surface radiation $Q^*$ and is spent on warming the coarse-blocky AL ($H_{al}^\theta$ of $10-20\,\mathrm{MJ\,m^{-2}}$) and transmitted into the permafrost body beneath ($Q_{\mathrm{PF}}$ of $5-15\,\mathrm{MJ\,m^{-2}}$). The remaining energy $\mathrm{dev}_{al}$ of $52-94\,\mathrm{MJ\,m^{-2}}$ corresponds to a potential ice melt of $\mathrm{dev}_{al}/L_m$ of $160-280\,\mathrm{mm}$ w.e.. The date of snow melt-out and onset of thaw-season primarily explains the $\sim 40\%$

larger cumulative heat uptake in 2022 compared to 2021, and the heat uptake scales with the positive degree day sum (PDD). The direct ground-ice melt estimate $Q_m$ from the ablation measurements (Fig. 10), here converted to a heat flux via Eq. 11, tends to be larger than $\mathrm{dev}_{al}$ ('AP53' in Fig. 12), but agrees well at the end of the thaw season, i.e., the estimates of the total ice melt during the thaw season are consistent. Conspicuous 'warming spikes', i.e. rapid AL warming and downward heat fluxes, occurred in winter–spring and are caused by non-conductive heat transfer. Certainly during the spring snowmelt beneath a

thick, closed/insulating snow cover (Amschwand et al., 2024a), but possibly also during intra-winter melt events under a thin, discontinuous snow cover, these 'warming spikes' are caused by latent heat effects from refreezing of infiltrated snowmelt rather than air convection ('wind pumping', Humlum (1997); Juliussen and Humlum (2008)).

The impact of advective heat transfer by the rainwater increases with depth. In the AL, the warming effect of the rain heat flux is limited compared to the other heat fluxes (radiation, air convection) and overcompensated by the evaporative cooling effect and decreased insolation from the cloud cover during precipitation (Amschwand et al., 2024a). Also, the water content in the coarse material remains low and does not affect thermal properties. The effect of drip water on the heat flux plate measurement were not significant for daily to monthly energy budgets as shown by the good correlation to the pyrgeometer measurements (Amschwand et al., 2024b). However, in the permafrost body beneath the active layer, heat fluxes are smaller and the effects of rainwater (heat advection, changing thermal and mechanical properties) are not negligible, but hard to quantify with our measurements located in the uppermost 3 m.

## 5  Discussion

### 5.1  Summary of hydro-chemical findings

Many of the hydrological and isotope results on Murtèl rock glacier are along the lines of past studies and are briefly summarised here. A consistent finding are seasonal trends of discharge (decreasing), EC (increasing), and $\delta^{18}$O (increasing) interrupted by precipitation-related excursions (spikes of high discharge, low EC, and 'heavier' $\delta^{18}$O). This pattern is observed in both summers 2021 and 2022, despite different weather conditions. The synthesis plot EC–$\delta^{18}$O–Q–t (Fig. 13) plots the water samples of the Murtèl outflow as a function of EC, $\delta^{18}$O, discharge $Q$, and timing $t$. The plot suggests three end-member components whose contribution varies throughout the summer, namely: (1) Snowmelt (depleted isotopic composition, low EC, discharge high during weeks) dominant in early summer, (2) rainwater (enriched $\delta^{18}$O, low EC, discharge episodically high after rainfall) dominant after snowmelt, and (3) groundwater baseflow ('reacted groundwater' of intermediate $\delta^{18}$O, moderate–high EC, very low discharge) to which the system tends to in late summer–autumn (Aug–Oct; $S_E$ then stagnant with discharge beneath the detection threshold of 3 L min$^{-1}$). The plot is based on samples from spring $S_E$, which is the last to fall dry and provided the most complete data set, extended into autumn 2022 by the then discovered seep $Q^*$. The other main spring $S_W$ is different enough from nearby $S_E$ to hint at different water flow paths (a level of detail beyond the scope of this study), yet similar enough to provide a comparable picture. The Y-shape with three end members based on the three-component model using dual chemical (EC) and isotopic ($\delta^{18}$O) tracers agrees with previous studies on intact rock glaciers (Krainer and Mostler, 2002; Harrington et al., 2018; Winkler et al., 2021a). Snowmelt and rock glacier core/permafrost ice (Haeberli, 1990) have a similar isotopic fingerprint ($\delta^{18}$O of $-13$‰ to $-17$‰), i.e. using isotopes alone, the ground-ice meltwater is likely indistinguishable from snowmelt. Evidence for ground-ice melt is potentially available during dry phases after snowmelt is completed and its meltwater largely flushed out of the supra-permafrost aquifer, otherwise the signal is masked by snowmelt or diluted by rainwater. We suspect that the two strikingly depleted mid-July 2022 supra-permafrost samples collected during a dry spell–heat wave might most closely represent ground-ice melt ($-12.9$ and $-13.5$‰, Fig. 5). $S_E$ surface outflow at that moment was so small that the carefully placed EC logger was not submerged (data gap), the outflow infiltrated on the spot. Also, the $S_E$ outflow was diluted with enriched rainwater, as shown by the $\delta^{18}O$ of the manually sampled $S_E$ water ($-11.4$‰). The

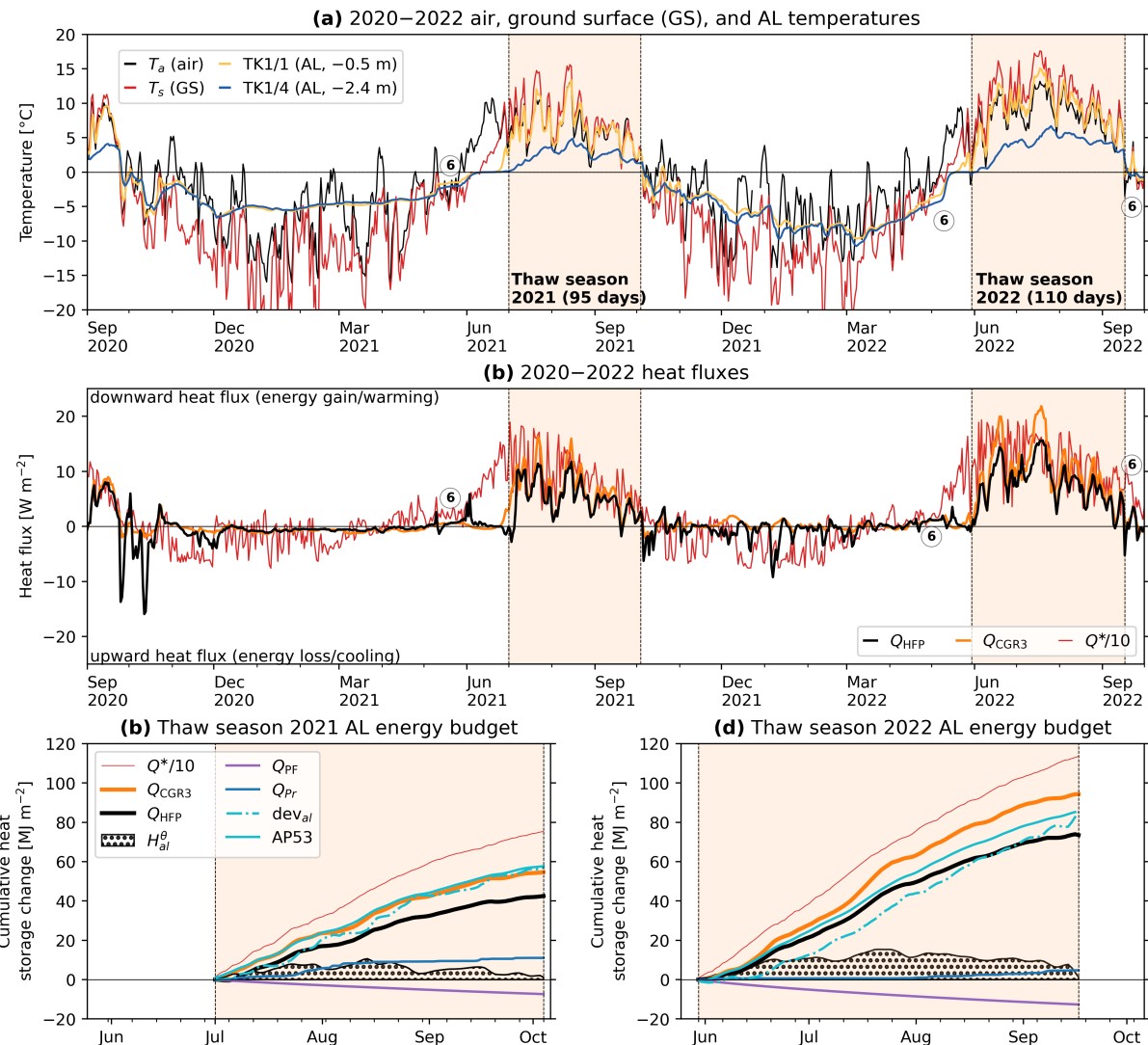

**Figure 12.** Measured **(a)** temperatures and **(b)** heat fluxes in the coarse-blocky Murtèl AL. 'Warming spikes' ⑥ indicate refreezing events. **(c–d)** Cumulative heat fluxes in the thaw season **(c)** 2021 and **(d)** 2022.

presumed signal of ground-ice melt recorded in the active layer is already lost in the rock-glacier spring, and associated surface outflow is very small ($\ll 3$ L min$^{-1}$).

## 5.2 The Murtèl plot-scale water balance

With our comprehensive hydro-meteorological data set, we resolve the water fluxes and ice storage changes on Murtèl rock glacier on plot scale for the hydrological years 2021–2022 (Table 3; Fig. 9), except for groundwater (discussed in Sect. 5.4). The plot scale refers to the point-wise (not areal) ablation observations, precipitation gauging and measurements for the AL

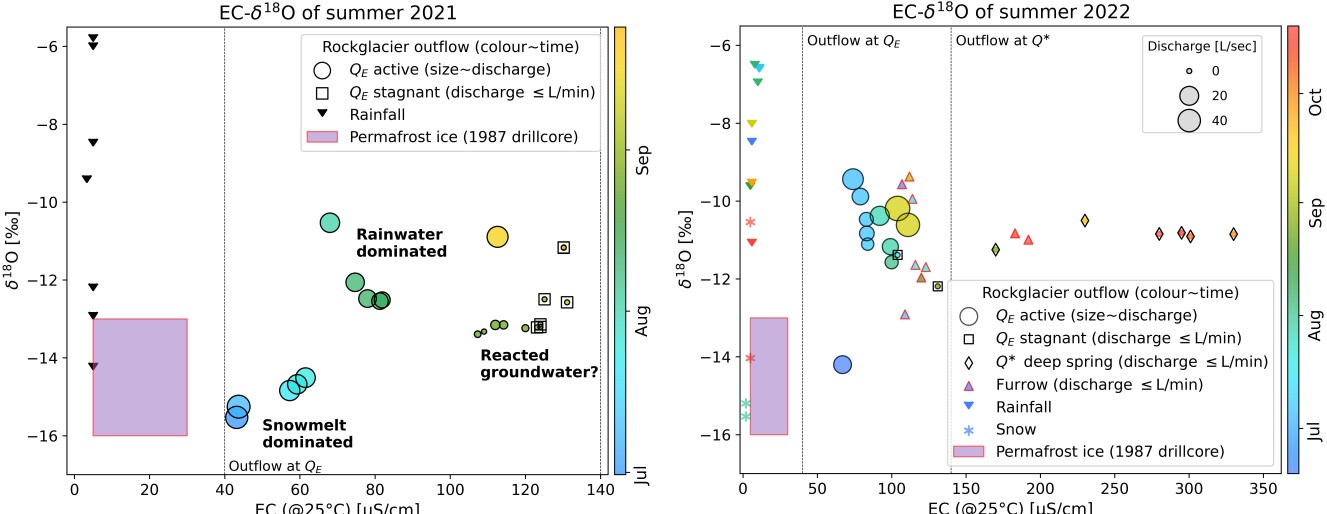

**Figure 13.** Synthesis: EC–$\delta^{18}$O–Q–t plot showing the inter-annual variability of the two summers 2021 and 2022 with different weather (main spring $S_E$ and seep $S^*$). Since EC increases throughout the summer, the EC axis can be roughly read as time axis (time/season is shown by the colours).

energy budget on the rock glacier. The rock glacier outflow is normalised by the catchment area (30 ha) inferred from the topography, i.e. the rock glacier and the surrounding talus and rock faces. The thaw-season water budgets are closed, but annual water budgets are not. During the thaw season after near-completion of the snowmelt (Jul–Sep 2021 and Jun–Sep 2022), surface stream discharge equals available precipitation (i.e., precipitation minus evaporation) and there is no sustained baseflow,

suggesting that liquid water storage changes are negligible (within the precision of the water budget) and that little rainwater infiltrates (Eq. 1) in Arnoux et al., 2020) (these assumptions are revisited in Sect. 5.4). Discharge during the snowmelt period (Apr–Jun 2021 and Apr–May 2022) and consequently the annual discharge are strongly underestimated because discharge before snow melt-out of the forefield is not gauged.

    The EC and stable isotope signature of the Murtèl surface outflow suggest that there must be some liquid water storage in

the catchment which is however too small to sustain surface baseflow over more than a few days and is not resolved by the water balance. Evidence for aquifers come from the seasonal increase in in EC (a proxy for mineralisation) and the $\delta^{18}$O of the outflow which is buffered to the rainfall $\delta^{18}$O (Figs. 4–5, 13). In particular, the late-summer water sampled in the 'deep seep' $S^*$ (Fig. 2) which shows a stable $\delta^{18}$O value of $-10.9 \pm 0.2\,‰$ (Fig. 5d), intermediate between depleted snowmelt (around $-15‰$) and enriched summer rainfall (around $-8‰$), might hint at a sub-permafrost aquifer in the overdeepened bedrock

depression (Fig. 14b) (Arenson et al., 2010) recharged from both snowmelt (or icemelt) and rainfall and a water residence time of several months.

    The single largest contribution comes from the snowmelt which amounts to 65 % of the total annual precipitation released in a few weeks during spring freshet (Table 3; Figs. 4, 5). This is typical for seasonally snow-covered mountain sites. Here, we

**Table 3.** Murtèl rock glacier plot-scale water balance in terms of cumulative water fluxes for the respective seasons: winter (Oct–Mar), spring snow melt until end surface zero curtain (May–Jun/Jul), thaw season/summer–autumn (Jun/Jul–Sep), and annual.

| Cumulative water flux | Hydrological year 2020–2021 | | | | Hydrological year 2021–2022 | | | |
|---|---|---|---|---|---|---|---|---|
| [mm w.e.] | Oct–Mar | Apr–Jun | Jul–Sep | *annual* | Oct–Mar | Apr–May | Jun–Sep | *annual* |
| SWE (dev$_{\mathrm{SEB}}$) | — | 915 | $\sim 0^f$ | *915* | — | 596 | 2 | *598* |
| Rainfall (on-site/underline{corrected}) | 23$^g$ | 0$^g$ | 462/502 | *485/525* | 0$^g$ | 0$^g$ | >163/319 | *163/319* |
| Evaporation and sublimation (SEB) | 33 | 24 | <89$^h$ | *<146* | 74 | 17 | <117$^h$ | *<208* |
| Available precipitation$^a$ | | | 373/413 | *1254/1294* | | | 46/204 | *551/709* |
| Watershed area-normalised outflow$^b$ | 39 | >140$^i$ | 415 | *>595* | 0 | >129$^i$ | 206 | *>336* |
| Ground-ice melt$^c$ | | | 167 | *167* | | | 261 | *261* |
| Energy budget dev$_{al}$/'ablation stake' $Q_m$ | 0 | 0 | 167/167 | | 0 | 0 | 257/265 | |
| *Ratios* | | | | | | | | |
| Water balance closure$^d$ | | | 100 % | *46 %* | | | 101 % | *47 %* |
| Snowpack retention ratio$^e$ | | | | *18%* | | | | *44%* |
| Ground ice melt / total precipitation | | | 33% | *12%* | | | 82% | *28%* |
| Ground ice melt / available precipitation | | | 40% | *13%* | | | 128% | *37%* |
| Ground ice melt / surface outflow | | | 40% | | | | 127% | |

$^a$ Available precipitation is rainfall and snowmelt (winter precipitation) minus evaporation/sublimation. $^b$Total surface outflow measured at gauging station (Fig. 2), normalized by catchment area of 30 ha (Fig. 2). $^c$Ground ice melt is the average of two independent estimates of ground ice storage changes (cf. Table 3 in Amschwand et al. (2024b)). $^d$Surface outflow / available precipitation. Deviation arising from infiltration and measurement errors (notably incompletely gauged snowmelt, water level–discharge relation, and uncertain catchment delineation/area). $^e$Amount of accreted ground ice (that is derived from the snowpack) divided by the SWE. $^f$ Surviving snow patches in catchment (Fig. 2) sustained some snowmelt throughout summer 2021, but it was too small to produce measurable surface outflow. $^g$Rain-on-snow events not quantifiable by our measurement setup. $^h$Evaporation likely upper bound. $^i$Outflow not gauged when water-level sensor beneath snow cover; i.e. no measurable discharge as long as the rock glacier field is snow covered.

focus on the contribution of the ground-ice melt compared to precipitation and outflow of the plot-scale water balance (Table 3, Fig. 9). First, in the hot–dry year 2022 with a snow-poor winter, accumulation and melt in coarse-blocky AL seasonally stored and released up to $44\%$ of the winter precipitation (SWE; "snowpack retention ratio" in Table 3) and $28\%$ of the annual precipitation ($37\,\%$ of the available precipitation if evaporation/sublimation is subtracted) in form of temporarily fixed ground ice. In the cool–moist year 2021 and a winter with average snowfall, the relative importance of the ground-ice melt was smaller: $18\,\%$ of SWE, $12\,\%$ of the yearly precipitation ($13\,\%$ of the available precipitation). All these components show a strong, weather-sensitive inter-annual variability. Second, negative ground ice storage changes amount to $127\%$ of the surface outflow during the hot–dry summer 2022 ($40\,\%$ in the cool, rainy summer 2021), but the meltwater does not appear as surface outflow: The streambed dries out in phases without precipitation (Sect. 5.4). However, even if all of the icemelt reached the surface springs (if the permafrost body were impervious), meltwater exposed to the atmosphere outside of the protecting AL would have a local importance only. Evaporation rates are similarly high as ice melt rates, the meltwater would practically evaporate on the spot (Fig. 9).

## 5.3 Substantial ground ice turnover in the AL, little meltwater from degrading permafrost

Most ($90-100\,\%$) of the generated (ice) meltwater is derived from winter precipitation accumulated in the AL (active-layer thaw), not from the ice-rich, but quasi-inert permafrost body (rock glacier core) beneath.

### 5.3.1 Seasonal ice turnover in the AL

We obtained consistent estimates of seasonal ground ice melt from the AL energy budget ($\mathrm{dev}_{al}$, Fig. 12) and ablation observations ($Q_m$, Figs. 9, 10), although we estimated the AL energy budget $\mathrm{dev}_{al}$ beneath a broad ridge and made ablation observations $Q_m$ in a narrow furrow. End-of-thaw season values (cumulative ice melt) are $160\,\mathrm{mm}$ w.e. in the thaw season 2021, and $260\,\mathrm{mm}$ w.e. in 2022 (Table 3). Although our point-wise observations cannot exclude some differential accumulation/ablation in furrows and ridges (a micro-topographic variability mentioned by Kääb et al. (1998) and Halla et al. (2021)), the agreement suggests that our estimates of end-of-thaw season ice storage changes are fairly representative over the landform and within an uncertainty of $\pm 50\,\%$ (Fig. 11). Smaller discrepancies in the sub-seasonal evolution of $Q_m$ and $\mathrm{dev}_{al}$ (Fig. 12) likely arise from differences in the micro-topographic setting, duration of snow cover, debris texture, and AL thickness at the two measurement points. Our estimates of the icemelt agree with prior studies on Murtèl by Scherler et al. (2014) and Pruessner et al. (2022).

The 'ablation stake' measurements show no local AL thickening for the years 2021–2023 (Fig. 10). The ground ice that melted during the thaw season was regenerated by refreezing and accumulation in the following winter and spring. In the exceptionally cool–wet summer 2021 with late snow melt-out (begin of thaw season in July), even net accumulation occurred at least locally in the rock-glacier furrow (frozen thermistor TK4/5; Fig. 4a). The surviving snow patches (Fig. 2) tentatively support the net positive mass balance observed in 2021, but cannot provide conclusive evidence on a landform scale.

A net increase in below-ground ice content during snowmelt, i.e. the conversion of snowmelt to ground ice, as observed on Murtèl is arguably most efficient on well-drained coarse-blocky permafrost landforms. Such terrain, abundant in periglacial high-mountain areas, features a distinct seasonal chain of coupled heat and water–ice transformations co-controlled by the snow cover (illustrated in Fig. 14a (1)–(3)). In autumn–early winter, the permeable AL contains little water to freeze, enabling a rapid and pervasive ground cooling to large depths before the onset of an insulating snowcover (Renette et al., 2023; Luetschg et al., 2008) (Fig. 14a (1)). In late winter and during spring snowmelt, whenever a warm and melting snowpack releases water into the subfreezing AL, the large cold content (sensible heat) is partly transformed into the build-up of new ground ice (latent heat) (Fig. 14a (2); 'warming spikes' in Fig. 12). The timing of ground ice build-up, *from snowmelt in spring* rather than by freezing soil water in autumn–early winter (Mendoza López et al., 2024), is distinct from fine-grained material with a larger water retention capacity (Renette et al., 2023) and is observed in other coarse-blocky permafrost landforms (e.g., Sawada et al., 2003; Rist and Phillips, 2005; Marchenko et al., 2012, 2024). From a hydraulic perspective, the large and permeable pores provide ample storage space for the infiltrating snowmelt to refreeze and in-blown snow to be trapped *at depth* without blocking the water infiltration/percolation pathways by the newly formed ground ice (Woo, 2012) or basal ice at the ground–snow interface (Phillips et al., 2017) (but see also Scherler et al. (2010)). During the thaw season, the melting AL

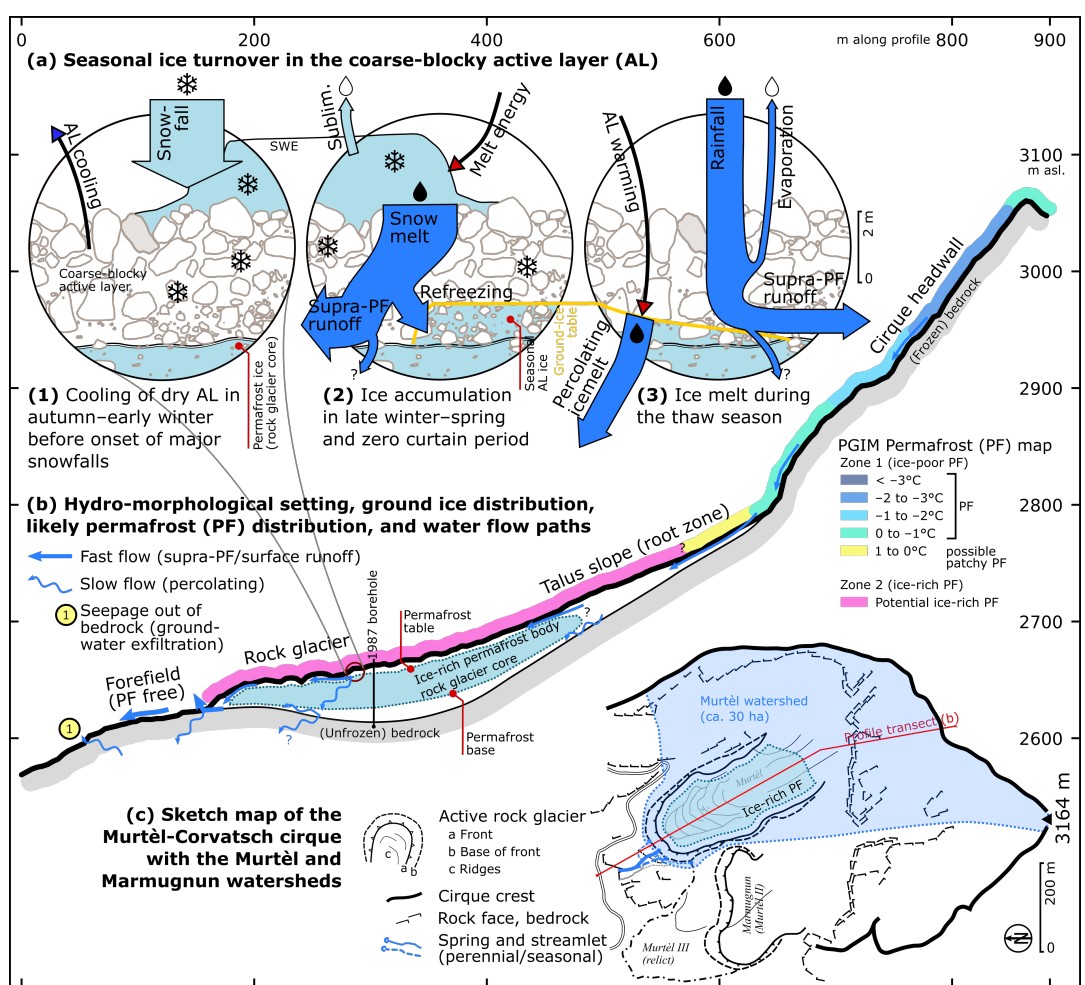

**Figure 14.** Conceptual model of the near-surface hydrogeology of the periglacial Murtèl watershed. **(a)** Seasonal thermo-hydrological processes in the Murtèl coarse-blocky AL. Water fluxes (thick arrows) for the 2021/22 hydrological year (arrows to scale, Table 3). **(b)** Profile through the Murtèl watershed. Rock glacier structure drawn after Vonder Mühll (1993); PGIM permafrost distribution after Kenner et al. (2019). **(c)** Sketch map of the Murtèl watershed in its cirque.

ice largely ($\sim 80\,\%$) absorbs the ground heat flux (Fig. 14a (3); Amschwand et al. (2024b)). Numerical simulations by Renette
et al. (2023) of the coupled heat and water/ice balance apply to Murtèl. They showed that alone this interplay between the
subsurface water/ice balance and ground freezing/thawing leads to negative ground thermal anomalies (up to $2.2\,°C$ in their
modelling scenarios), i.e. even without air convection which is an additional efficient cooling mechanism common for such
terrain (Amschwand et al., 2024b; Wicky and Hauck, 2020; Wicky et al., 2024; Amschwand et al., 2024b). The regeneration of
ground ice in the AL partly explains the climate resilience of coarse-blocky landforms (Scherler et al., 2013; Amschwand et al.,
2024b). If the lost ground ice is not regenerated, the permafrost landform is preconditioned towards irreversible degradation

(Hilbich et al., 2008; Hauck and Hilbich, 2024). Finally, permafrost conditions might be required in seasonally snow-covered terrain to keep winter ground temperatures well below the freezing point until the spring snowmelt, depending on the timing and insulation of the snow cover. This is shown by the 'bottom temperature of snow cover' (BTS temperatures) (Haeberli, 1973, 1975; Haeberli and Patzelt, 1982), a winter-equilibrium ground surface temperature attained beneath a closed/insulating snow cover, that are near or above the freezing point in permafrost-free (seasonally frozen) Alpine terrain. To sum up, ground ice accumulation by snowmelt refreezing relies on strong (preferentially convective) ground cooling and water supply into the subfreezing AL, in turn sensitive to debris texture, thermo-hydraulic properties, and weather/snow conditions.

### 5.3.2    Melt of permafrost ice in the rock glacier core

In our warming climate, net accumulation as possibly in 2021 is the exception. Murtèl rock glacier is slowly degrading (AL thickening (Noetzli et al., 2019)) and must be on multi-year average releasing meltwater from the 'old' permafrost core. Neither our ablation observations nor our energy balance measurements that span only two years can reliably separate melt from superimposed 'young' AL ice from melt of 'old' permafrost ice (and a possible 'transient layer' in between, Mendoza López et al. (2024)). Inter-annual to decadal surface subsidence estimates from kinematic surveys (Barsch and Hell, 1975; Kääb et al., 1998; Müller et al., 2014; Müller et al., 2016) and vertical borehole deformation (Arenson et al., 2002) revealed slow subsidence attributed to ground ice melt in the over-saturated permafrost body. Kääb et al. (1998) report $2-5$ cm yr$^{-1}$ for the period 1987–1996. The most recent 2022–2023 geodetic measurements (I. Gärtner-Roer, pers. comm.) yield $1$ cm yr$^{-1}$ in the mid–frontal parts of the rock glacier (no significant net change in the surface elevation: the ice melt compensates compressive thickening; subsidence $\approx$ ice loss in massive ice). The amount of meltwater released from the permafrost core is an order of magnitude smaller than the seasonal ice turnover in the AL and negligible compared to precipitation, surface outflow, and even evaporation, agreeing with previous studies (Bearzot et al., 2023).

### 5.4    Near-surface hydrology of the periglacial Murtèl watershed: Meteoric water runs off, icemelt infiltrates

The periglacial Murtèl watershed underlain by either ice-rich permafrost (active rock glacier and talus slope) or steep bedrock (Fig. 14b, c) has a small storage and retention capacity for liquid water (cf. Geiger et al., 2014; Rogger et al., 2017). The thaw-season hydrograph is flashy, responding rapidly (little delay) and strongly (high peak discharge) to daily oscillating snowmelt and rainfall events (Figs. 4, 5). There is no sustained baseflow during winter and summer droughts ($\lesssim 3$ L min$^{-1}$ for a 30 ha watershed). Once snowmelt is completed, the rock glacier springs fall dry a few days after the last rainfalls. Surface outflow is derived from snowmelt and precipitation (meteoric water, Fig. 14a (2)–(3)), as shown by the isotopic signature and the thaw-season water balance (Table 3). Importantly, we could neither observe surface runoff from ground-ice melt nor retrieve a clear isotopic signal in the rock-glacier springs during hot–dry weather spells (Figs. 5d, 13b), although extrapolating the melt rates over the ice-underlain rock-glacier area ($\sim 150 \times 300$ m$^2$ à 3 mm w.e. day$^{-1}$) would yield $\sim 100$ L min$^{-1}$. Whereas snowmelt and rainwater leave the watershed largely as supra-permafrost runoff (some unknown fraction does infiltrate; Fig. 14a (2)–(3)), no traces of the ice melt are visible (except perhaps on the rock glacier itself, Fig. 13b).

Whereas comparatively intense rainfall that exceeds the infiltration capacity largely runs off as supra-permafrost water and generates surface streamflow, slowly generated ice melt (limited by the small ground heat flux $Q_G$) fully infiltrates into the permafrost body. In fact, observed and calculated ground-ice melt rates of $1-4$ mm w.e. day$^{-1}$ ($\sim 10^{-8}$ m s$^{-1}$; Fig. 9) are beneath the surface runoff threshold or infiltration capacity of $9$ mm day$^{-1}$ indicated by the rainfall–streamflow relation (Fig. 7). Tracer tests by Tenthorey and Gerber (1991) proved a sub-surface hydraulic connection to the front of the relict Murtèl III rock glacier (Figs. 2, A1). Also the Murtèl drilling campaigns showed that neither the permafrost body nor the unfrozen bedrock beneath are impermeable: Supra-, intra- and sub-permafrost water flow in conduit-like taliks, water inflow, and air loss during drilling operations are reported in Vonder Mühll and Haeberli (1990); Vonder Mühll (1992) and Arenson et al. (2002, 2010). Also Halla et al. (2021) estimated that $58-89\%$ of the seasonal meltwater left the *Dos Lenguas* rock glacier hydrologic system via groundwater pathways. Cicoira et al. (2019) and Kenner et al. (2020) inferred rapid percolation of snowmelt through the permafrost body to the basal shear zone based on the widely observed spring-time rock glacier acceleration immediately after onset of the snowmelt.

The Murtèl periglacial watershed is not a 'teflon basin' (cf. Williams et al., 2015). In mountain permafrost, the ground ice table (permafrost table) is a semi-impervious aquitard that restricts (but not completely prevents) vertical water flow. Mountain permafrost is inherently discontinuous and characterized by a mosaic distribution of perennially and seasonally cryotic ground with varying ground temperatures and water/ice content that reflects the variable micro-climatic conditions in the complex terrain (illustrated in Fig. 14b). Zones of "warm" permafrost (containing unfrozen water) and taliks enhance the hydraulic permeability (connectivity) of discontinuous permafrost and allow for water movement and groundwater recharge/discharge (Cheng and Jin, 2012; Luethi et al., 2017; Wagner et al., 2020; Arenson et al., 2022; Bearzot et al., 2023). We propose that icemelt in the abundant coarse-blocky permafrost landforms – intact rock glaciers, frozen talus slopes, block fields – contribute to groundwater recharge despite low melt rates, because melt rates are steadily sustained over the entire thaw season (insulating effect of the AL), the infiltration efficiency is high, and the infiltrating icemelt is additional to the infiltrating snowmelt and rainwater (Fig. 14a). As glaciers retreat, permafrost thaws and snowpacks diminish in the rapidly deglacierizing mountains, groundwater as the most climatically resilient freshwater resource will be increasingly important to sustain baseflow in the headwaters of rivers originating in mountain ranges (Hayashi, 2020). Additionally, the cold groundwater might sustain 'icy seeps' that are climate refugia for cold-adapted aquatic organisms (Hotaling et al., 2019; Tronstad et al., 2020; Brighenti et al., 2019a, b, 2021; Bearzot et al., 2023). Since coarse-blocky permafrost landforms are abundant in periglacial high-mountain areas, understanding their climate-controlled future changes, quantifying subsurface water/ice storage and pathways in mountain permafrost-underlain catchments (Woo, 2011; Louis et al., 2024), and assessing the cryosphere–groundwater connectivity are policy-relevant research topics (cf. van Tiel et al., 2024).

## 6 Conclusions

Rock glaciers store and release water and ice on different time scales with varying magnitudes and residence times: (1) Liquid water on short-term (sub-monthly) scale, (2) ground ice in the coarse-blocky AL on intermediate term (seasonal), and (3) 'old'

permafrost ice on long-term (over millennia). We quantify supra-permafrost ice storage changes, precipitation, evaporation and snowmelt on the intact Murtèl rock glacier (Engadine, eastern Swiss Alps) and the surface outflow of its periglacial, permafrost-underlain watershed. The single-lobe rock glacier ($4\,\text{ha}$) and its watershed ($30\,\text{ha}$) are small, unglacierized, and sparsely vegetated. Boreholes revealed a rock glacier stratigraphy of $2-5\,\text{m}$ thick coarse-blocky AL over $\sim 30\,\text{m}$ of nearly massive ice (perennially frozen rock glacier core). Our comprehensive hydro-meteorological measurements resolve the water balance and snow/ice storage changes between ground surface and permafrost table at plot scale, i.e. isolated on the rock glacier itself. We estimated ice storage changes (melt and accumulation) in the coarse-blocky active layer (AL) from direct observations of the seasonal evolution of the ground-ice table ('ablation stake' measurements) and an AL energy budget derived from in-situ heat flux measurements.

The thaw-season hydrograph at the rock-glacier front is flashy with rapidly varying discharge typical for a small, permafrost-underlain catchment with limited water storage capacity: Precipitation peaks are followed by dry phases without baseflow when the rock glacier springs temporarily fall dry. The isotopic signature and electrical conductivity of the outflow suggests that the Murtèl surface outflow is largely derived from snowmelt and rainwater. No clear hydro-chemical evidence ($\delta^{18}\text{O}$ natural tracer) of permafrost meltwater could be found in the surface outflow. In contrast, the in-situ ablation measurements and AL energy budget concurrently suggested substantial seasonal ground ice turnover in coarse blocky AL with accumulation and melt of $150-300\,\text{mm w.e.}$ ($160\,\text{mm w.e.}$ in 2021, $260\,\text{mm w.e.}$ in 2022). $20-40\,\%$ of the snowpack is accumulated directly (in-blown snow) or refreezes onto the convectively cooled blocks as snowmelt infiltrates into the coarse blocky AL during winter warm spells and the spring snowmelt, forming annually replenished superimposed ground ice. This freeze/thaw storage protracts the snowmelt into late summer and absorbs $\sim 80\,\%$ of the thaw-season ground heat flux, i.e. is a coupled thermo-hydrological buffer that releases meltwater in late summer and protects the underlying permafrost body, contributing to the rock glaciers' climate resilience. Consequently, the amount of meltwater released from the 'old' permafrost ice (long-term storage in the rock glacier core) is currently $\sim 10\,\text{mm w.e. yr}^{-1}$, an order of magnitude smaller than the AL meltwater contribution and negligible compared to the inter-annual variations of precipitation. Melting superimposed ice sourced from winter precipitation cannot increase the total annual runoff. We hypothesize that the generated meltwater (which we could not track in the surface outflow) largely percolates deeper and recharges the sub-permafrost groundwater. Ground ice melt rates are limited by the ground heat flux to $1-4\,\text{mm w.e. day}^{-1}$, which is below the $9\,\text{mm day}^{-1}$ infiltration capacity inferred from the local rainfall–streamflow relation. Sub-surface water pathways have been identified by tracer tests (Tenthorey and Gerber, 1991).

Our case study supports the concept that intact rock glacier sustain baseflow by recharging groundwater with meltwater from the seasonal AL thaw, in addition to the large, climate-resilient sub-surface permafrost ice reserves they hold (Jones et al., 2019; Hayashi, 2020; Wagner et al., 2021; Arenson et al., 2022; Navarro et al., 2023). Specific to permeable, coarse blocky landforms with a small water retention capacity as Murtèl rock glacier, little water is stored in liquid form. Instead, the water is immobilized by forming ice in the AL: Snowmelt is rapidly fixed as AL ice in winter–spring, but slowly released during the thaw season, routing a small, but sustained meltwater flow through the low-permeability permafrost aquitard to deeper aquifers. The superimposed AL ice is refrozen winter precipitation and cannot increase the total annual runoff. While our

isolated energy and water balance investigations provided detailed and quantitative insights into hydro-thermal processes in the Murtèl AL, integrated hydrogeological studies are needed to address rock glaciers as embedded in their watersheds, namely the connectivity to adjoining landforms and the groundwater, sub-surface groundwater pathways, and their capacity to sustain baseflow compared to other high-mountain landforms such as relict rock glaciers, moraines, meadows, and wetlands.

715 *Data availability.* The PERMOS data can be obtained from the PERMOS network (http://www.permos.ch), and the PERMA-XT measurement data from https://www.permos.ch//doi/permos-spec-2023-1 (doi:10.13093/permos-spec-2023-01).

# Appendix A: Additional figures

## A1 Hydro-morphological sketch map

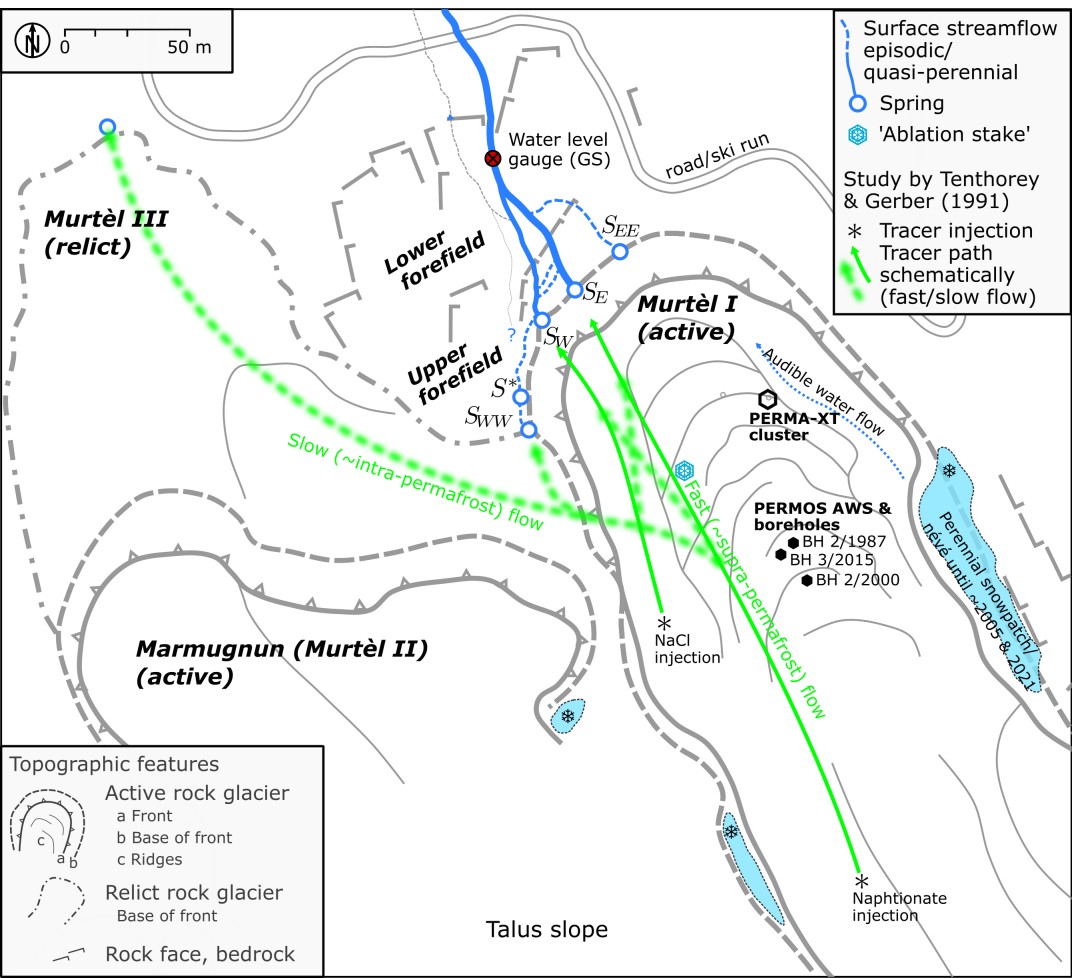

**Figure A1.** Hydro-morphological setting. Sketch map of the active rock glaciers Murtèl I and Marmugnun (Murtèl II), and the relict rock glacier Murtèl III.

## A2 Field observation: Six snapshots of the strongly variable discharge

**(a) Situation end June** (30.6)
Discharge >50 L/sec (dilution gauging).

**(b) Situation early-mid August** (10–12.8)
Discharge ~5–10 L/sec (dilution gauging).

**(c) Situation end August** (23–27.8)
Discharge ~5–10 L/min (bucket method).
Rapid recession to <3 L/min until Aug 27.

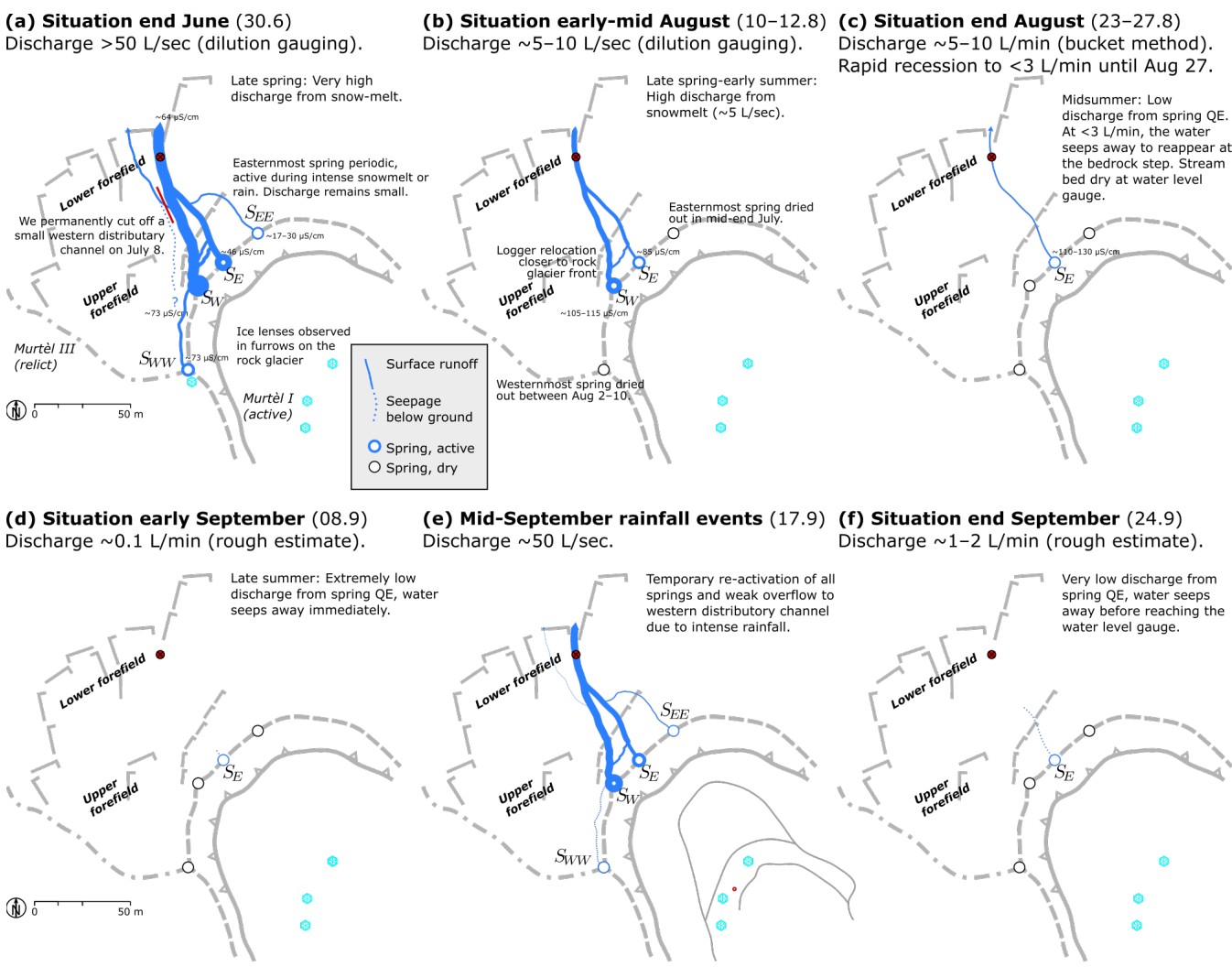

**(d) Situation early September** (08.9)
Discharge ~0.1 L/min (rough estimate).

**(e) Mid-September rainfall events** (17.9)
Discharge ~50 L/sec.

**(f) Situation end September** (24.9)
Discharge ~1–2 L/min (rough estimate).

**Figure A2.** Six snapshots of the strongly variable discharge on the rock glacier forefield in summer 2021 as observed on the regular field visits. **(a)–(c)**: Spring–early summer (end June until end August) with declining discharge as snowmelt progressed; **(d)–(f)**: Late summer–autumn (end August until end September) with strong discharge changes within days as a response to rainfall events. The baseflow in dry periods seeps into the ground before reaching the gauging station in the lower forefield.

 **A3    Rainfall MeteoSuisse station *Piz Corvatsch* (COV)**

Rainfall at the PERMA-XT station on Murtèl and on MeteoSuisse station *Piz Corvatsch* (actually located on a promontory of *Piz Murtèl*) is shown in Fig. A3.

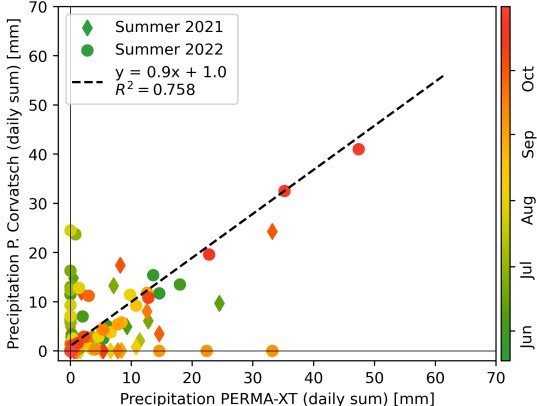

**Figure A3.**

## A4 Deuterium excess $d_{excess}$

Deuterium excess [‰] is defined as (Williams et al., 2006)

$$d_{excess} := \delta^2\mathrm{H} - 8 \cdot \delta^{18}\mathrm{O}. \tag{A1}$$

Results for the thaw season 2021 and 2022 are shown in Fig. A4.

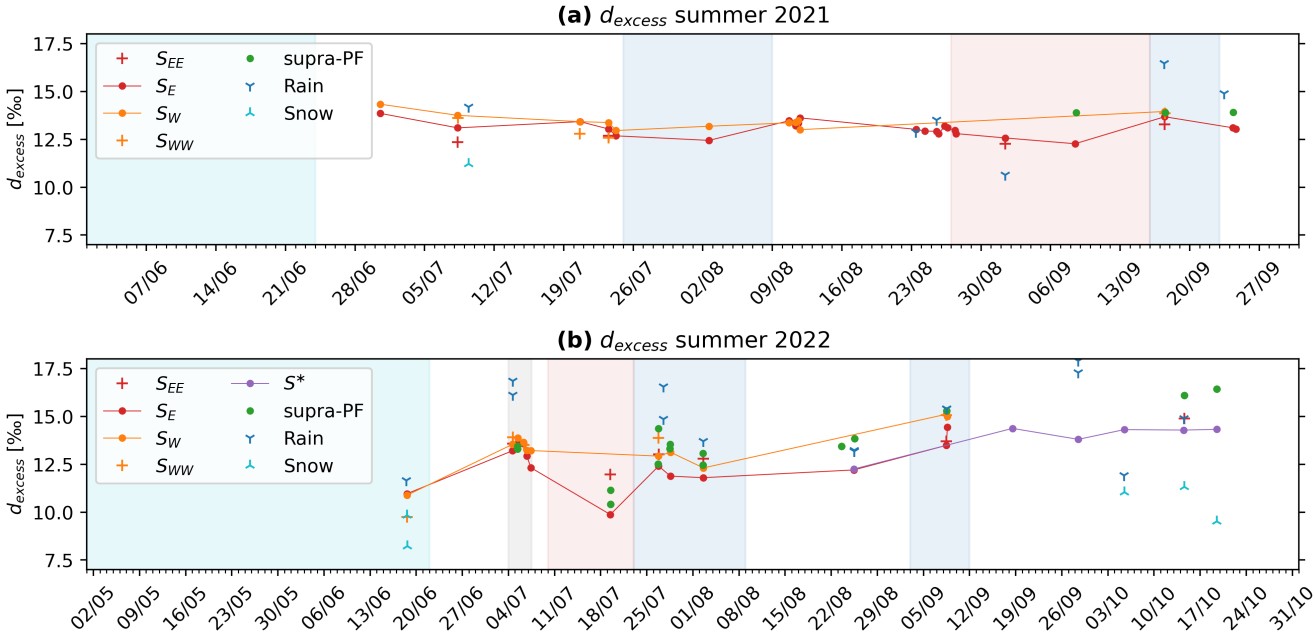

**Figure A4.** Deuterium excess $d_{excess}$ (Eq. A1) for the summers **(a)** 2021 and **(b)** 2022.

## Appendix B: Stable isotope data

The isotope data are listed in Tables B1–B5.

**Table B1.** Isotope data (1/5). Water temperature and EC measured in the field with WTW probe.

| ID | Job | Sampling date | Location | $T_w$ [°C] | EC [$\mu$S cm$^{-1}$] | $\delta^{18}$O [‰] | $\delta^2$H [‰] | $d_{excess}$ [‰] |
|---|---|---|---|---|---|---|---|---|
| ISO01 | 21-233 | 2020-08-27, 13:50 | SE | 1.2 | 122 | -13.37 | -95.67 | 11.29 |
| ISO02 | 21-233 | 2020-08-27, 14:00 | SW | 0.5 | 214 | -12.79 | -89.95 | 12.37 |
| ISO03 | 21-233 | 2021-06-30, 13:04 | SE | -0.1 | 43 | -15.53 | -110.38 | 13.86 |
| ISO04 | 21-233 | 2021-06-30, 13:07 | SW | 0.2 | 69 | -16.13 | -114.74 | 14.33 |
| ISO05 | 21-233 | 2021-07-08, 08:00 | SEE | 4.5 | 28 | -13.75 | -97.66 | 12.35 |
| ISO06 | 21-233 | 2021-07-08, 08:00 | SE | -0.2 | 44 | -15.25 | -108.93 | 13.10 |
| ISO07 | 21-233 | 2021-07-08, 08:00 | SW | -0.1 | 67 | -15.99 | -114.16 | 13.75 |
| ISO08 | 21-233 | 2021-07-08, 08:25 | SWW | -0.2 | 54 | -15.82 | -112.98 | 13.62 |
| ISO09 | 21-233 | 2021-07-08, 10:00 | rSE | 1.7 | 86 | -15.50 | -110.42 | 13.59 |
| ISO10 | 21-233 | 2021-07-09, 10:00 | RW | 3.8 | 3 | -9.48 | -61.68 | 14.20 |
| ISO11 | 21-233 | 2021-07-09, 10:00 | SS | 0.0 | 5 | -14.41 | -104.04 | 11.21 |
| ISO12 | 21-233 | 2021-07-20, 15:15 | SE | 0.1 | 57 | -14.84 | -105.29 | 13.43 |
| ISO13 | 21-233 | 2021-07-20, 15:05 | SW | 0.0 | 73 | -15.61 | -111.45 | 13.42 |
| ISO14 | 21-233 | 2021-07-20, 14:55 | SWW | 0.1 | 43 | -15.44 | -110.72 | 12.80 |
| ISO15 | 21-233 | 2021-07-23, 12:30 | SEE | 1.5 | 66 | -12.57 | -87.84 | 12.68 |
| ISO16 | 21-233 | 2021-07-23, 12:30 | SE | 0.1 | 62 | -14.51 | -103.03 | 13.04 |
| ISO17 | 21-233 | 2021-07-23, 12:30 | SW | -0.1 | 73 | -15.47 | -110.37 | 13.38 |
| ISO18 | 21-233 | 2021-07-23, 12:30 | SWW | 0.1 | 54 | -14.88 | -106.46 | 12.57 |
| ISO19 | 21-233 | 2021-07-23, 12:30 | rSE | 4.5 | 91 | -15.06 | -107.81 | 12.68 |
| ISO20 | 21-233 | 2021-07-24, 06:30 | SE | 0.1 | 59 | -14.68 | -104.75 | 12.68 |
| ISO21 | 21-233 | 2021-07-24, 06:30 | SW | -0.1 | 74 | -15.42 | -110.44 | 12.96 |
| ISO22 | 21-233 | 2021-08-02, 15:00 | SE | 0.1 | 68 | -10.53 | -71.80 | 12.44 |
| ISO23 | 21-233 | 2021-08-02, 14:50 | SW | 0.0 | 92 | -10.68 | -72.28 | 13.18 |
| ISO24 | 21-233 | 2021-08-10, 15:56 | SE | 0.2 | 75 | -12.06 | -83.01 | 13.47 |
| ISO25 | 21-233 | 2021-08-10, 15:56 | SW | 0.0 | 111 | -12.30 | -85.02 | 13.36 |
| ISO26 | 21-233 | 2021-08-11, 08:00 | SE | 0.0 | 82 | -12.51 | -86.89 | 13.20 |
| ISO27 | 21-233 | 2021-08-11, 08:05 | SW | 0.0 | 98 | -12.70 | -88.25 | 13.38 |
| ISO28 | 21-233 | 2021-08-11, 15:03 | SE | 0.4 | 81 | -12.54 | -87 | 13.35 |
| ISO29 | 21-233 | 2021-08-11, 15:11 | SW | 0.3 | 111 | -12.68 | -87.99 | 13.47 |
| ISO30 | 21-233 | 2021-08-11, 11:37 | BR | 15.5 | 33 | -9.51 | -67.29 | 8.76 |
| ISO31 | 21-233 | 2021-08-11, 18:48 | SE | 0.0 | 78 | -12.48 | -86.20 | 13.62 |
| ISO32 | 21-233 | 2021-08-11, 18:54 | SW | 0.0 | 98 | -13.12 | -91.96 | 13.00 |

Analytical uncertainty: 0.1‰ for $\delta^{18}$O, 1.5‰ for $\delta^2$H ($1\sigma$). Cf. Fig. 2 for sampling locations. Sample location abbreviated as follows: RW rainwater (RWe event, sampled at every field visit; RWi ca. monthly integrated). SS snow sample. SEE easternmost main spring $S_{EE}$. SE eastern main spring $S_E$. SW Murtèl western main spring $S_W$. SWW westernmost main spring $S_{WW}$. SD deep seep $S^*$. rSE seep in front of relict Murtèl III rock glacier. sPF supra-permafrost water sampled in rock-glacier furrow (sPFc next to ablation stake, sPFt next to thermistor TK4/5). BR bedrock seepage (Fig. 2).

**Table B2.** Isotope data (2/5). Water temperature and EC measured in the field with WTW probe.

| ID | Job | Sampling date | Location | $T_w$ [°C] | EC [µS cm$^{-1}$] | $\delta^{18}$O [‰] | $\delta^2$H [‰] | $d_{excess}$ [‰] |
|---|---|---|---|---|---|---|---|---|
| ISO33 | 21-258 | 2021-08-23, 11:00 | RW | | 5 | -5.86 | -33.97 | 12.89 |
| ISO34 | 21-258 | 2021-08-23, 11:40 | SE | 0.5 | 107 | -13.39 | -94.12 | 13.02 |
| ISO35 | 21-258 | 2021-08-24, 08:33 | SE | 0.5 | 109 | -13.33 | -93.67 | 12.93 |
| ISO36 | 21-258 | 2021-08-25, 12:35 | SE | | 112 | -13.15 | -92.32 | 12.92 |
| ISO37 | 21-258 | 2021-08-25, 17:55 | SE | | 114 | -13.15 | -92.43 | 12.80 |
| ISO38 | 21-258 | 2021-08-25, 12:30 | RW | | 5 | -12.99 | -90.45 | 13.51 |
| ISO39 | 21-258 | 2021-08-26, 08:10 | SE | | 120 | -13.24 | -92.72 | 13.19 |
| ISO40 | 21-258 | 2021-08-26, 15:40 | SE | | 123 | -13.22 | -92.63 | 13.11 |
| ISO41 | 21-258 | 2021-08-27, 09:00 | SE | 0.3 | 124 | -13.21 | -92.69 | 12.96 |
| ISO42 | 21-258 | 2021-08-27, 12:01 | SE | 0.5 | 124 | -13.14 | -92.33 | 12.80 |
| ISO43 | 21-258 | 2021-09-01, 11:00 | SEE | 3.0 | 85 | -12.07 | -84.32 | 12.27 |
| ISO44 | 21-258 | 2021-09-01, 11:00 | SE | 2.0 | 125 | -12.50 | -87.40 | 12.57 |
| ISO45 | 21-258 | 2021-09-01, 11:00 | RW | | 5 | -14.30 | -103.74 | 10.64 |
| ISO46 | 21-258 | 2021-09-08, 11:13 | SE | | 131 | -12.57 | -88.32 | 12.27 |
| ISO47 | 21-258 | 2021-09-08, 11:00 | RW | | 5 | -8.55 | -49.03 | 19.34 |
| ISO48 | 21-258 | 2021-09-08, 13:45 | sPFc | 0.0 | | -12.97 | -89.86 | 13.89 |
| ISO49 | 21-258 | 2021-09-17, 11:40 | SEE | | | -10.58 | -71.37 | 13.28 |
| ISO50 | 21-258 | 2021-09-17, 11:43 | SE | | 112 | -10.89 | -73.46 | 13.68 |
| ISO51 | 21-258 | 2021-09-17, 11:45 | SW | | 150 | -10.47 | -69.79 | 13.95 |
| ISO52 | 21-258 | 2021-09-17, 12:03 | SWW | | | -10.56 | -70.61 | 13.86 |
| ISO53 | 21-258 | 2021-09-17, 13:30 | sPFc | | | -10.56 | -70.55 | 13.90 |
| ISO55 | 21-258 | 2021-09-24, 08:20 | SE | | 130 | -11.16 | -76.22 | 13.10 |
| ISO56 | 21-258 | 2021-09-24, 15:24 | SE | | 130 | -11.17 | -76.33 | 13.04 |
| ISO57 | 21-258 | 2021-09-24, 09:00 | sPFc | | | -11.77 | -80.22 | 13.91 |
| ISO58 | 21-258 | 2021-09-23, 11:35 | RW | | 5 | -12.26 | -83.21 | 14.89 |
| ISO59 | 21-258 | 2021-09-17, 11:00 | RW | | 5 | -6.07 | -32.10 | 16.46 |

Cf. Table B1 for abbreviations of sample locations.

**Table B3.** Isotope data (3/5). Water temperature and EC measured in the field with WTW probe.

| ID | Job | Sampling date | Location | $T_w$ [°C] | EC [µS cm$^{-1}$] | $\delta^{18}$O [‰] | $\delta^2$H [‰] | $d_{excess}$ [‰] |
|---|---|---|---|---|---|---|---|---|
| ISO2201 | 22-139 | 2022-04-20, 12:00 | SS | -2.0 | 3 | -18.83 | -139.12 | 11.51 |
| ISO2202 | 22-139 | 2022-04-20, 12:20 | SS | -2.0 | 2 | -20.32 | -151.27 | 11.32 |
| ISO2203 | 22-139 | 2022-05-20, 12:30 | GS | -0.2 | 68 | -17.10 | -124.43 | 12.39 |
| ISO2204 | 22-139 | 2022-06-18, 14:09 | SE | 0.0 | 67 | -14.20 | -102.66 | 10.96 |
| ISO2205 | 22-136 | 2022-06-18, 14:13 | SW | -0.2 | 85 | -14.83 | -107.75 | 10.88 |
| ISO2206 | 22-139 | 2022-06-18, 14:20 | SWW | 0.0 | 41 | -14.84 | -109.01 | 9.75 |
| ISO2207 | 22-136 | 2022-06-18, 12:40 | RWe | 22.7 | 6 | -8.55 | -56.73 | 11.65 |
| ISO2208 | 22-139 | 2022-06-18, 15:30 | SS | 0.0 | 2 | -15.20 | -113.35 | 8.23 |
| ISO2209 | 22-139 | 2022-06-18, 15:20 | SS | 0.0 | 2 | -15.53 | -114.43 | 9.83 |
| ISO2210 | 22-139 | 2022-07-04, 15:55 | SEE | 7.9 | 48 | -8.92 | -57.75 | 13.57 |
| ISO2211 | 22-136 | 2022-07-04, 15:25 | SE | -0.1 | 74 | -9.44 | -62.30 | 13.21 |
| ISO2212 | 22-136 | 2022-07-04, 15:30 | SW | 0.1 | 98 | -9.46 | -62.15 | 13.55 |
| ISO2213 | 22-139 | 2022-07-04, 15:35 | SWW | -0.1 | 90 | -10.32 | -68.63 | 13.92 |
| ISO2214 | 22-139 | 2022-07-04, 16:30 | RWe | 16.6 | 11 | -6.65 | -37.09 | 16.12 |
| ISO2215 | 22-139 | 2022-07-04, 16:30 | RWi | 16.6 | 11 | -6.68 | -36.56 | 16.86 |
| ISO2216 | 22-136 | 2022-07-05, 10:00 | SE | -0.2 | 79 | -9.88 | -65.49 | 13.58 |
| ISO2217 | 22-136 | 2022-07-05, 10:05 | SW | -0.1 | 110 | -10.02 | -66.32 | 13.87 |
| ISO2218 | 22-136 | 2022-07-05, 09:30 | sPFt | 0.2 | 78 | -9.69 | -64.03 | 13.45 |
| ISO2219 | 22-136 | 2022-07-05, 09:00 | sPFc | 0.0 | 107 | -9.57 | -63.22 | 13.31 |
| ISO2220 | 22-139 | 2022-07-06, 07:35 | SE | -0.1 | 83 | -10.47 | -70.19 | 13.57 |
| ISO2221 | 22-139 | 2022-07-06, 07:40 | SW | -0.1 | 116 | -10.61 | -71.24 | 13.66 |
| ISO2222 | 22-139 | 2022-07-06, 07:50 | SWW | 0.1 | 95 | -10.68 | -71.89 | 13.51 |
| ISO2223 | 22-139 | 2022-07-06, 19:30 | SE | | 83 | -10.83 | -73.72 | 12.93 |
| ISO2224 | 22-139 | 2022-07-06, 19:35 | SW | | | -10.87 | -73.72 | 13.22 |
| ISO2225 | 22-139 | 2022-07-06, 19:10 | sPFc | | | -10.77 | -73.21 | 12.95 |
| ISO2226 | 22-139 | 2022-07-07, 09:25 | SE | | 83 | -11.11 | -76.53 | 12.33 |
| ISO2227 | 22-139 | 2022-07-07, 09:35 | SW | | | -10.96 | -74.43 | 13.22 |
| ISO2228 | 22-139 | 2022-07-07, 08:40 | sPFc | | | -10.99 | -75.13 | 12.76 |
| ISO2229 | 22-139 | 2022-07-19, 11:30 | SEE | 12.0 | 68 | -11.86 | -82.91 | 11.97 |
| ISO2230 | 22-136 | 2022-07-19, 11:20 | SE | 4.2 | 104 | -11.39 | -81.21 | 9.87 |
| ISO2231 | 22-136 | 2022-07-19, 15:00 | RWe | | | -4.79 | -35.41 | 2.93 |
| ISO2232 | 22-139 | 2022-07-19, 12:25 | sPFt | 1.9 | 60 | -13.39 | -95.94 | 11.14 |
| ISO2233 | 22-136 | 2022-07-19, 12:40 | sPFc | 0.0 | 109 | -12.91 | -92.87 | 10.41 |

Cf. Table B1 for abbreviations of sample locations.

**Table B4.** Isotope data (4/5). Water temperature and EC measured in the field with WTW probe.

| ID | Job | Sampling date | Location | $T_w$ [°C] | EC [µS cm$^{-1}$] | $\delta^{18}$O [‰] | $\delta^2$H [‰] | $d_{excess}$ [‰] |
|---|---|---|---|---|---|---|---|---|
| ISO2234 | 22-139 | 2022-07-26, 19:30 | SEE | 5.1 | 70 | -9.93 | -66.41 | 13.03 |
| ISO2235 | 22-136 | 2022-07-26, 19:10 | SE | -0.1 | 92 | -10.38 | -70.63 | 12.40 |
| ISO2236 | 22-136 | 2022-07-26, 19:05 | SW | 0.0 | 124 | -9.72 | -64.80 | 12.93 |
| ISO2237 | 22-139 | 2022-07-26, 18:55 | SWW | -0.1 | 103 | -10.06 | -66.64 | 13.87 |
| ISO2238 | 22-139 | 2022-07-26, 18:30 | sPFt | 0.0 | 85 | -10.21 | -67.29 | 14.37 |
| ISO2239 | 22-136 | 2022-07-26, 18:00 | sPFc | 0.0 | 114 | -9.94 | -67.02 | 12.52 |
| ISO2240 | 22-139 | 2022-07-27, 12:50 | RWe | | 10 | -7.02 | -39.64 | 16.55 |
| ISO2241 | 22-136 | 2022-07-27, 12:50 | RWi | | 8 | -6.57 | -37.73 | 14.85 |
| ISO2242 | 22-136 | 2022-07-28, 14:30 | SE | | 99 | -11.17 | -77.49 | 11.88 |
| ISO2243 | 22-136 | 2022-07-28, 14:30 | SW | 5.0 | 142 | -10.57 | -71.39 | 13.13 |
| ISO2244 | 22-139 | 2022-07-28, 13:45 | sPFt | 0.0 | 95 | -11.61 | -79.36 | 13.54 |
| ISO2245 | 22-139 | 2022-07-28, 12:36 | sPFc | 0.0 | 116 | -11.64 | -79.81 | 13.31 |
| ISO2246 | 22-139 | 2022-08-02, 14:10 | SEE | 6.3 | 75 | -10.38 | -70.28 | 12.79 |
| ISO2247 | 22-136 | 2022-08-02, 14:20 | SE | 0.4 | 100 | -11.57 | -80.77 | 11.80 |
| ISO2248 | 22-136 | 2022-08-02, 14:25 | SW | 6.4 | 161 | -10.52 | -71.83 | 12.31 |
| ISO2249 | 22-139 | 2022-08-02, 14:30 | RWe | 23. | 5 | -9.69 | -63.81 | 13.71 |
| ISO2250 | 22-139 | 2022-08-02, 13:10 | sPFt | 0.0 | 96 | -11.91 | -82.25 | 13.07 |
| ISO2251 | 22-136 | 2022-08-02, 12:35 | sPFc | 0.0 | 123 | -11.69 | -81.09 | 12.46 |
| ISO2252 | 22-139 | 2022-08-25, 10:30 | SE | 4.0 | 131 | -12.19 | -85.32 | 12.20 |
| ISO2253 | 22-136 | 2022-08-25, 11:00 | SD | 0.2 | 170 | -11.25 | -77.75 | 12.26 |
| ISO2254 | 22-136 | 2022-08-25, 11:30 | RWe | 19.5 | 6 | -8.09 | -51.45 | 13.23 |
| ISO2255 | 22-139 | 2022-08-25, 11:30 | RWi | 19.5 | 6 | -8.37 | -53.81 | 13.17 |
| ISO2256 | 22-136 | 2022-08-25, 13:00 | sPFc | 0.8 | 120 | -11.96 | -81.87 | 13.84 |
| ISO2257 | 22-139 | 2022-08-23, 14:00 | sPFc | | | -11.52 | -78.75 | 13.44 |
| ISO2258 | 22-139 | 2022-09-08, 11:10 | SEE | 1.7 | 51 | -9.32 | -60.85 | 13.71 |
| ISO2259 | 22-136 | 2022-09-08, 11:15 | SE | 0.1 | 104 | -10.19 | -68.04 | 13.49 |
| ISO2260 | 22-136 | 2022-09-08, 11:20 | SW | 0.3 | 164 | -9.59 | -61.65 | 15.11 |
| ISO2261 | 22-139 | 2022-09-08, 11:25 | SWW | -0.1 | 131 | -9.62 | -61.90 | 15.06 |
| ISO2262 | 22-139 | 2022-09-08, 12:45 | RWe | 12.0 | 6 | -9.65 | -61.78 | 15.40 |
| ISO2263 | 22-136 | 2022-09-08, 12:45 | RWi | 12.0 | 6 | -9.60 | -61.41 | 15.40 |
| ISO2264 | 22-136 | 2022-09-08, 12:20 | sPFc | 0.2 | 112 | -9.37 | -59.68 | 15.29 |
| ISO2265 | 22-136 | 2022-09-08, 15:30 | SE | -0.0 | 111 | -10.61 | -70.46 | 14.44 |
| ISO2266 | 22-136 | 2022-09-08, 15:35 | SW | 0.0 | 163 | -9.50 | -61.04 | 14.99 |

Cf. Table B1 for abbreviations of sample locations.

**Table B5.** Isotope data (5/5). Water temperature and EC measured in the field with WTW probe.

| ID | Job | Sampling date | Location | $T_w$ [°C] | EC [µS cm$^{-1}$] | $\delta^{18}$O [‰] | $\delta^{2}$H [‰] | $d_{excess}$ [‰] |
|---|---|---|---|---|---|---|---|---|
| ISO2267 | 22-136 | 2022-09-18, 12:35 | SD | 0.1 | 230 | -10.50 | -69.63 | 14.38 |
| ISO2268 | 22-136 | 2022-09-28, 12:00 | SD | -0.1 | 330 | -10.85 | -72.99 | 13.80 |
| ISO2269 | 22-139 | 2022-09-28, 11:30 | RWe | 0.0 | 10 | -11.70 | -75.69 | 17.90 |
| ISO2270 | 22-136 | 2022-09-28, 11:30 | RWi | 0.0 | 6 | -11.14 | -71.83 | 17.30 |
| ISO2271 | 22-136 | 2022-10-05, 11:50 | SD | -0.1 | 301 | -10.91 | -72.96 | 14.31 |
| ISO2272 | 22-139 | 2022-10-05, 11:30 | RWe | | 5 | -15.56 | -112.59 | 11.91 |
| ISO2273 | 22-136 | 2022-10-05, 12:00 | SS | 0.0 | 5 | -14.03 | -101.22 | 11.05 |
| ISO2274 | 22-136 | 2022-10-14, 14:05 | SEE | 1.8 | 92 | -10.78 | -71.31 | 14.90 |
| ISO2275 | 22-136 | 2022-10-14, 12:30 | SD | 0.0 | 295 | -10.82 | -72.25 | 14.29 |
| ISO2276 | 22-139 | 2022-10-14, 13:40 | RWe | | 5 | -12.49 | -85.05 | 14.89 |
| ISO2277 | 22-136 | 2022-10-14, 14:55 | sPFc | 0.0 | 183 | -10.83 | -70.54 | 16.09 |
| ISO2278 | 22-136 | 2022-10-14, 14:55 | SS | 0.0 | 5 | -10.54 | -72.99 | 11.33 |
| ISO2279 | 22-136 | 2022-10-19, 13:15 | SD | 0.0 | 280 | -10.85 | -72.44 | 14.32 |
| ISO2280 | 22-136 | 2022-10-19, 12:25 | sPFc | 0.0 | 192 | -10.99 | -71.50 | 16.43 |
| ISO2281 | 22-139 | 2022-10-19, 12:25 | SS | 0.0 | 6 | -11.33 | -81.14 | 9.52 |

Cf. Table B1 for abbreviations of sample locations.

*Author contributions.* DA performed the fieldwork, model development and analyses for the study and wrote the manuscript. MS, MH and
730 BK supervised the study, provided financial and field support and contributed to the manuscript preparation. AH and CK provided logistical
support and editorial suggestions on the manuscript. HG designed the novel sensor array, regularly checked data quality, contributed to the
analyses and provided editorial suggestions on the manuscript.

*Competing interests.* The authors declare that they have no conflict of interest.

*Acknowledgements.* This work is a collaboration between the University of Fribourg and GEOTEST and was funded by the Swiss Innovation
Agency Innosuisse (project 36242.1 IP-EE 'Permafrost Meltwater Assessment eXpert Tool PERMA-XT'). The authors wish to thank Walter
Jäger (Waljag GmbH, Malans) and Thomas Sarbach (Sarbach Mechanik, St. Niklaus) for the technical support, Stephan Bolay (GEOTEST)
for the field assistance (dilution gaugings), and the Corvatsch cable car company for logistical support. Insightful discussions with Theo
Jenk (Paul Scherrer Institute PSI), Isabelle Gärtner-Roer and Andreas Vieli (University of Zurich), Landon Halloran and Clément Roques
(University of Neuchâtel) contributed to the manuscript.

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
