# Peer review of "Seasonal ice storage changes and meltwater generation at Murtèl rock glacier (Engadine, eastern Swiss Alps): Estimates from measurements and energy budgets in the coarse blocky active layer"

_EGUsphere, 2024_

## Referee Comment (RC1)

**Review on HESS manuscript egusphere-2024-844: "On the hydrological significance of rock glaciers: A case study from Murtèl rock glacier (Engadine, eastern Swiss Alps) using below-ground energy-flux measurements, ground-ice melt observations and hydrological measurements"**

The authors provide a comprehensive summary of the outstanding field experiments and previous hydrological research conducted at the Murtèl rock glacier. This study is certainly of interest to the hydrology readership. However, I have some major concerns:

**Major Concerns**:

1. Could you provide more details on the influence of precipitation in this research? For example, how does precipitation affect the ground heat flux measurements and what impact does it have on the below-ground water/ice content?

2. Ice content is generally more challenging to determine than water content in field experiments. How do the authors measure the ice content?

3. Section 2.2 provides an introduction to past hydrological and hydro-chemical investigations. While recognizing past work is important, this section is somewhat lengthy and detracts readers from the focus of the current research. Simplifying this section and focusing more on the authors' field experiments, for example, combining Section 2.1 and Section 2.2 together into Section 2 'Study Site', would improve the manuscript.

4. Although the area has been extensively studied in the past decades, and it is good to recognize others work, it seems that the manuscript spends too much space on reviewing others' work. For example, in the result section, there are a lot of sentences stating that some observations are consistent with some previous study, but this will make reader distracted from the

innovative points of this research. It might be better to move some of others work in results and discussion part to introduction, and distinguish the key unique findings in this research.

5. In Sections 3.2.3 and 4.3, the authors discuss latent heat flux and evaporation. Have the authors considered the role of sublimation, particularly when there is snow coverage, in the turbulent heat flux discussion?

**Minor Suggestions**:

1. The abbreviation 'EC' first appears in line 71, but its explanation is only provided in line 212. Please introduce the abbreviation upon its first occurrence. Similar to 'EC', the term 'SEB', which I guess is the surface energy balance, also has the similar problem. It first appears in Line 252, but surface energy balance first appears in Line 151. And please also check the upper/lower case together. For example, in Line 151, it appears that the sentence should be: The surface energy balance (SEB) and AL-internal heat fluxes towards......

2. In line 269, $Q_{LE}$ should refer to latent rather than sensible heat flux.

3. There is no need to add a subheading in line 361 (4.2.1 Field observations). The paragraph can directly follow line 360.

4. Some sentences are repeated, please check. For example, the sentence in the caption of Figure 2 is mentioned again in lines 349-350.

---

## Author Comment (AC1)

Dear editor Hongkai Gao, Dear reviewers

We thank the two reviewers for their constructive, careful and encouraging reviews. In the following, we would like to reply to their comments and concerns.

We agree with reviewer #2, Ryan Webb, that we have not invented a new conceptual hydrological model of rock glaciers. The three different time scales of water/ice fluxes (short-, intermediate-, and long-term storage) that we refer to is established knowledge (e.g., Jones et al., 2019). Rather, our novel contribution regards the quantification of each flux at Murtèl rock glacier based on an unprecedently comprehensive data set, notably of the direct measurement of the seasonal ground ice accumulation/melt in the active layer and its connection to the surface energy balance. We also agree it is a local case study but emphasize that the calculations of the energy and water budgets are in our opinion far beyond established knowledge for a ventilated coarse-blocky active layer and include novel sensor technology. They are outlined in two publications (Amschwand et al., 2023, 2024) that this manuscript builds on. In the revised manuscript, we will reframe the storyline by rephrasing notably the introduction (introducing the existing conceptual rock glacier model more clearly) and the conclusions, more clearly separating our own results from others' contribution (including the proposed additional review publications). We will shorten the review of others' work in our result and discussion sections. With these changes we also respond to the same issues raised by reviewer #1. We propose as a new title: "Seasonal ice storage changes and meltwater generation at Murtèl rock glacier (Engadine, eastern Swiss Alps): Estimates from measurements and energy budgets in the coarse blocky active layer".

Minor points listed by line number (reviewer comment in blue, our response in black):
15: this 13% of annual precipitation and outflow is odd to me. Could you separate precip. And outflow somehow? Maybe " ~13% of annual precipitation becomes refrozen and ground ice that melts to become outflow"... or something similar.
Your remark made us aware of a perhaps confusing use of terminology: In the current draft, "ground ice melt" refers to both the ice melt in the active layer (on the rock glacier) and the meltwater in the surface outflow at the rock glacier front, although these are two distinct quantities estimated at different locations with different methods. To make things clearer, 'ground ice storage change' will specifically denote the accumulation and melt of ground ice as observed/estimated in the active layer from the 'ablation measurements' and the energy budget. The distinction from 'ground ice melt' is important because the ground ice storage change largely infiltrates and could not actually be measured as meltwater or 'ground ice melt' in the surface outflow.

19: The 50 mm is not 10 times smaller than the 150-300 mm values mentioned earlier. Am I mis-reading this?
This should be understood as an order of magnitude. We will improve this part by providing better estimates in the revised version and will improve the wording. On the well investigated Murtèl rock glacier, estimates of the permafrost ice melt (that is additional to the active layer ice melt) might be available.

111-119: This paragraph seems to be more fit in the study site description.
Agreed. The introduction will be rewritten.

Section 2.2 seems a bit too long and could be shortened.
We will shorten the discussion of the accessible literature (also raised by reviewer #1). However, we will still provide in this paragraph some comments on site-specific literature which is not easily accessible, namely Haeberli (1990), Tenthorey and Gerber (1991), and Stucki (1995).

211-213: This doesn't seem like a complete sentence, let along a stand-alone paragraph.

Yes, we'll delete this paragraph.

234: "CHECK!!" Please proof the manuscript for typos and missing references.
Yes, we'll carefully check this occurrence and also the entire manuscript.

342: Why are you only analyzing 2 years? Much of these instruments have been in place since 2009 correct? Why not a longer analysis? Perhaps listing what years some of these instruments were installed could clarify this.
Heat flux estimates and hydrological data are available from September 2020 to September 2023, but isotope measurements are only available from the summers 2021 and 2022 (only the plastic tube mentioned in L314 was drilled in 2009 but not used until our investigation). We will state the installation dates of the sensors more clearly.

Fig. 4 & 5: what are the horizontal dashed lines in Discharge and Temp.?
In temperature (panel a), it is the precipitation-outflow threshold (Fig.7), in discharge (panel b) it is the range covered by the gaugings. We'll update the figures accordingly.

369: The 8 gaugings seemed to happen mostly in the lower discharges. I would recommend trying to get some higher flows as well to better constrain the stage-Q relation.
Yes, we are aware of the shortcomings of the stage-discharge relation and mention that in the manuscript. However, we do not think that more gaugings add substantial value to this study. The focus of this study is on late-summer low flow where the ice melt storage changes potentially contribute most to the outflow. The absence of surface outflow during droughts, which is an important piece for our hypothesis that the meltwater preferentially infiltrates, is reliably measured.

384: "is" should be "are"
Yes, we'll modify the text accordingly.

531-532: this last sentence seems odd and out of place.
We'll delete the sentence.

596-597: similar comment before about being more clear for the "outflow and precipitation"
We rephrase the sentence to convey the message that these ratios of ground ice meltwater and precipitation refer to the ground ice storage changes. The meltwater from ground ice could not be actually measured as surface outflow (the conundrum, L605). To avoid the confusion, we'll modify Table 3 and rename "ground-ice melt" to "ground ice storage changes", as already mentioned above.

603-604: OK, makes sense. But, how much of the core is melting? I know this is really difficult to obtain, but some estimate could really help for any aquifer recharge or groundwater resources estimate to complete the hydrologic relevance of the paper.
Murtèl rock glacier is intensely investigated by several research groups. While no recent (<5 yr) estimates of subsidence or permafrost ice melt on Murtèl are published to our knowledge, we are positive that some more information could be gathered.

616-617: This all makes complete sense, but I think you should also discuss the future need to determine the subsurface pathways for these sources.
Yes, permafrost-groundwater interactions are an underinvestigated topic in mountain permafrost. We'll briefly discuss this, referring to a recently published preprint by Louis et al. (2024) on the hydrogeology of rock glaciers.
Response to reviewer #1 (reviewer comment in blue, our response in black):

1. Could you provide more details on the influence of precipitation in this research? For example, how does precipitation affect the ground heat flux measurements and what impact does it have on the below-ground water/ice content?

The effect of drip water on the heat flux plate measurement were not significant for daily to monthly energy budgets as shown by the good correlation to the pyrgeometer measurements. The coarse blocks are well drained. We chose not to reiterate these technical aspects for the sake of brevity and refer to Amschwand et al. (2024). In contrast, we agree that the impact of rainwater on ground ice melt deserves to be briefly discussed in this manuscript. It will be along these lines:

- In the active layer, where the bulk of the meltwater is generated, the warming effect of the rain heat flux is limited compared to the other heat fluxes (radiation, air convection) and overcompensated by the evaporative cooling effect and decreased insolation from the cloud cover during precipitation (Amschwand et al., 2023). Also, water content in the coarse material remains low and does not affect thermal properties.
- In the permafrost body beneath the active layer, where heat fluxes are small, the effects of rainwater (heat advection, changing thermal properties) are not negligible, but hard to quantify with our measurements in the uppermost 3 meters of the active layer. We discuss this more carefully, but nonetheless think that it does not detract from our conclusions because the meltwater contribution from the permafrost body is small.

2. Ice content is generally more challenging to determine than water content in field experiments. How do the authors measure the ice content?

The ice content is estimated from measurements of the ground ice table in the rock glacier furrow and an estimated porosity of the block mantle. We could not measure porosity in the coarse-blocky active layer. Instead, we will include a porosity as an additional parameter in the uncertainty estimate (Fig. 11).

3. Section 2.2 provides an introduction to past hydrological and hydro-chemical investigations. While recognizing past work is important, this section is somewhat lengthy and detracts readers from the focus of the current research. Simplifying this section and focusing more on the authors' field experiments, for example, combining Section 2.1 and Section 2.2 together into Section 2 'Study Site', would improve the manuscript.

We agree that our manuscript would benefit from shortening the discussion of others' works. However, in this introductory Sect. 2.2, we want to summarize site-specific past studies which are not easily accessible (see answer to reviewer #2 above). Instead, we'll shorten the reviews of others' work in our result and discussion section:

4. Although the area has been extensively studied in the past decades, and it is good to recognize others work, it seems that the manuscript spends too much space on reviewing others' work. For example, in the result section, there are a lot of sentences stating that some observations are consistent with some previous study, but this will make reader distracted from the innovative points of this research. It might be better to move some of others work in results and discussion part to introduction, and distinguish the key unique findings in this research.

In Sections 3.2.3 and 4.3, the authors discuss latent heat flux and evaporation. Have the authors considered the role of sublimation, particularly when there is snow coverage, in the turbulent heat flux discussion?

Yes, sublimation is an important latent heat flux when the ground is snow covered. It is included in the surface energy balance model and QLE (cf. Amschwand et al., 2023). We'll point that out in the revised manuscript.

The minor suggestions are:

1. The abbreviation 'EC' first appears in line 71, but its explanation is only provided in line 212. Please introduce the abbreviation upon its first occurrence. Similar to 'EC', the term 'SEB', which I guess is the surface energy balance, also has the similar problem. It first appears in Line 252, but surface energy balance first appears in Line 151. And please also check the upper/lower case together. For example, in Line 151, it appears that the sentence should be: The surface energy balance (SEB) and AL-internal heat fluxes towards......

We agree and modify the manuscript accordingly.

2. In line 269, QLE should refer to latent rather than sensible heat flux.

Yes, thank you for spotting this error. The sentence should read: "The evaporative water flux is derived from the latent turbulent flux QLE..."

3. There is no need to add a subheading in line 361 (4.2.1 Field observations). The paragraph can directly follow line 360.

We agree and modify the manuscript accordingly.

4. Some sentences are repeated, please check. For example, the sentence in the caption of Figure 2 is mentioned again in lines 349-350.

We'll remove repeated sentences for brevity.

REFERENCES

Amschwand, D., Scherler, M., Hoelzle, M., Krummenacher, B., Haberkorn, A., Kienholz, C., and Gubler, H.: Surface heat fluxes at coarse blocky Murtèl rock glacier (Engadine, eastern Swiss Alps), The Cryosphere Discussion, https://doi.org/10.5194/egusphere-2023-2109, 2023. (This publication has been accepted in the meantime)

Amschwand, D.,Wicky, J., Scherler, M., Hoelzle,M., Krummenacher, B., Haberkorn, A., Kienholz, C., and Gubler, H.: Sub-surface processes and heat fluxes at coarse-blocky Murtèl rock glacier (Engadine, eastern Swiss Alps), in review at Earth Surface Dynamics, 2024.

Haeberli, W., ed.: Pilot analysis of permafrost cores from the active rock glacier Murtèl I, Piz Corvatsch, Eastern Swiss Alps. A workshop report., no. 9 in Arbeitsheft, VAW/ETH Zürich, 1990.

Jones, D. B., Harrison, S., Anderson, K., and Whalley,W. B.: Rock glaciers and mountain hydrology: A review, Earth-Science Reviews, 193, 66–90, https://doi.org/10.1016/j.earscirev.2019.04.001, 2019.

Louis, C., Halloran, L. J. S., and Roques, C.: Seasonal and diurnal freeze-thaw dynamics of a rock glacier and their impacts on mixing and solute transport, EGUsphere [preprint], https://doi.org/10.5194/egusphere-2024-927, 2024.

Stucki, T.: Permafrosttemperaturen im Oberengadin. Eine Auswertung der Bohrlochtemperaturen im alpinen Permafrost des Oberengadins im Hinblick auf einen Erwärmungstrend und Schmelzwasserabfluss aus dem Permafrost., Master's thesis, Abteilung Erdwissenschaften der ETH Zürich, 1995.

Tenthorey, G. and Gerber, E.: Hydrologie du glacier rocheux de Murtèl (Grisons). Description et interprétation de traçages d'eau, in: Modèles en Géomorphologie – exemples Suisses. Session scientfique de la Société suisse de Géomorphologie. Fribourg, 22/23 juin 1990., edited by Monbaron, M. and Haeberli, W., vol. 3 of Rapport et recherches, Institut de Géographie Fribourg, 1991.

---

## Author Response (AR1)

Dear editor Hongkai Gao, Dear reviewers

We thank the two reviewers for their constructive, careful and encouraging reviews. In the following, we would like to outline how we have replied to their comments and concerns in the revised manuscript.
Aug 28, 2024

We agree with reviewer #2, Ryan Webb, that we have not invented a new conceptual hydrological model of rock glaciers. The three different time scales of water/ice fluxes (short-, intermediate-, and long-term storage) that we refer to is established knowledge (e.g., Jones et al., 2019). Rather, our novel contribution regards the quantification of each flux at Murtèl rock glacier based on an unprecedently comprehensive data set, notably of the direct measurement of the seasonal ground ice accumulation/melt in the active layer, its connection to the surface energy balance, and its connection to the groundwater. The latter is now the main message of the revised manuscript: We hypothesize that rock glaciers (and other coarse blocky permafrost landforms) recharge the groundwater, a hydrological role previously overlooked. Introduction and discussion have been completely rewritten, and an additional figure (Fig. 14) shows the seasonal thermo-hydrological processes in the Murtèl coarse blocky active layer. Other figures have been updated as well (Fig. 9 recalculated with the correct catchment area of 30 ha; Fig. 10 added 2024 data to show the 2023-2024 winter accumulation; Fig. 11 redrawn; Fig. 12 added panel (a) showing ground temperatures). The new title reads "Seasonal ice storage changes and meltwater generation at Murtèl rock glacier (Engadine, eastern Swiss Alps): Estimates from measurements and energy budgets in the coarse blocky active layer".

Minor points listed by line number (reviewer comment in blue, our response in black):
15: this 13% of annual precipitation and outflow is odd to me. Could you separate precip. And outflow somehow? Maybe " ~13% of annual precipitation becomes refrozen and ground ice that melts to become outflow"... or something similar.
This part has been revised.

19: The 50 mm is not 10 times smaller than the 150-300 mm values mentioned earlier. Am I mis-reading this?
This should be understood as an order of magnitude. We have improved this part by providing better estimates in the revised version.

111-119: This paragraph seems to be more fit in the study site description.
Agreed. The introduction has been rewritten.

Section 2.2 seems a bit too long and could be shortened.
We have shortene the discussion of the accessible literature (also raised by reviewer #1). However, we still provided in this paragraph some comments on site-specific literature which is not easily accessible, namely Haeberli (1990), Tenthorey and Gerber (1991), and Stucki (1995).

211-213: This doesn't seem like a complete sentence, let along a stand-alone paragraph.
Yes, we have deleted this paragraph.

234: "CHECK!!" Please proof the manuscript for typos and missing references.
Yes, we have carefully checked this occurrence and also the entire manuscript.

342: Why are you only analyzing 2 years? Much of these instruments have been in place since 2009 correct? Why not a longer analysis? Perhaps listing what years some of these instruments were installed could clarify this.

Heat flux estimates and hydrological data are available from September 2020 to September 2023, but isotope measurements are only available from the summers 2021 and 2022 (only the plastic tube mentioned in L314 was drilled in 2009 but not used until our investigation). We have stated the installation dates of the sensors more clearly.

Fig. 4 & 5: what are the horizontal dashed lines in Discharge and Temp.?
In temperature (panel a), it is the precipitation-outflow threshold (Fig.7), in discharge (panel b) it is the range covered by the gaugings. We have updated the figures or figure captions accordingly.

369: The 8 gaugings seemed to happen mostly in the lower discharges. I would recommend trying to get some higher flows as well to better constrain the stage-Q relation.
Yes, we are aware of the shortcomings of the stage-discharge relation and mention that in the manuscript. However, we do not think that more gaugings add substantial value to this study. The focus of this study is on late-summer low flow where the ice melt storage changes potentially contribute most to the outflow. The absence of surface outflow during droughts, which is an important piece for our hypothesis that the meltwater preferentially infiltrates, is reliably measured.

384: "is" should be "are"
Yes, we have modified the text accordingly.

531-532: this last sentence seems odd and out of place.
We have deleted the sentence.

596-597: similar comment before about being more clear for the "outflow and precipitation"
We rephrase the sentence to convey the message that these ratios of ground ice meltwater and precipitation refer to the ground ice storage changes.

603-604: OK, makes sense. But, how much of the core is melting? I know this is really difficult to obtain, but some estimate could really help for any aquifer recharge or groundwater resources estimate to complete the hydrologic relevance of the paper.
Murtèl rock glacier is intensely investigated by several research groups. While no recent (<5 yr) estimates of subsidence or permafrost ice melt on Murtèl are published to our knowledge, we could obtain the most recent subsidence measurements from I. Gärtner-Roer (University of Zurich).

616-617: This all makes complete sense, but I think you should also discuss the future need to determine the subsurface pathways for these sources.
Yes, permafrost-groundwater interactions are an underinvestigated topic in mountain permafrost. We have briefly discuss this, referring to a recently published preprint by Louis et al. (2024) on the hydrogeology of rock glaciers.

Response to reviewer #1 (reviewer comment in blue, our response in black):

1. Could you provide more details on the influence of precipitation in this research? For example, how does precipitation affect the ground heat flux measurements and what impact does it have on the below-ground water/ice content?

The effect of drip water on the heat flux plate measurement were not significant for daily to monthly energy budgets as shown by the good correlation to the pyrgeometer measurements. The coarse blocks are well drained. We chose not to reiterate these technical aspects for the sake of brevity and refer to Amschwand et al. (2024). In contrast, we agree that the impact of rainwater on ground ice melt deserves to be briefly discussed in this manuscript. It is along these lines (added in L503–510):

- In the active layer, where the bulk of the meltwater is generated, the warming effect of the rain heat flux is limited compared to the other heat fluxes (radiation, air convection) and overcompensated by the evaporative cooling effect and decreased insolation from the cloud cover during precipitation (Amschwand et al., 2023). Also, water content in the coarse material remains low and does not affect thermal properties.
- In the permafrost body beneath the active layer, where heat fluxes are small, the effects of rainwater (heat advection, changing thermal properties) are not negligible, but hard to quantify with our measurements in the uppermost 3 meters of the active layer. We discuss this more carefully, but nonetheless think that it does not detract from our conclusions because the meltwater contribution from the permafrost body is small.

2. Ice content is generally more challenging to determine than water content in field experiments. How do the authors measure the ice content?

The ice content is estimated from measurements of the ground ice table in the rock glacier furrow and an estimated porosity of the block mantle. We could not measure porosity in the coarse-blocky active layer. Instead, we have included porosity/ice content as an additional parameter in the uncertainty estimate (Fig. 11) with a conservative range of 0.2–0.8. It has not significantly affected the outcome.

3. Section 2.2 provides an introduction to past hydrological and hydro-chemical investigations. While recognizing past work is important, this section is somewhat lengthy and detracts readers from the focus of the current research. Simplifying this section and focusing more on the authors' field experiments, for example, combining Section 2.1 and Section 2.2 together into Section 2 'Study Site', would improve the manuscript.

We agree that our manuscript would benefit from shortening the discussion of others' works. However, in this introductory Sect. 2.2, we want to summarize site-specific past studies which are not easily accessible (see answer to reviewer #2 above). Instead, we have shortened the reviews of others' work in our result and discussion section:

4. Although the area has been extensively studied in the past decades, and it is good to recognize others work, it seems that the manuscript spends too much space on reviewing others' work. For example, in the result section, there are a lot of sentences stating that some observations are consistent with some previous study, but this will make reader distracted from the innovative points of this research. It might be better to move some of others work in results and discussion part to introduction, and distinguish the key unique findings in this research.

In Sections 3.2.3 and 4.3, the authors discuss latent heat flux and evaporation. Have the authors considered the role of sublimation, particularly when there is snow coverage, in the turbulent heat flux discussion?

Yes, sublimation is an important latent heat flux when the ground is snow covered. It is included in the surface energy balance model and QLE (cf. Amschwand et al., 2023). We have pointed that out in the revised manuscript.

The minor suggestions are:

1. The abbreviation 'EC' first appears in line 71, but its explanation is only provided in line 212. Please introduce the abbreviation upon its first occurrence. Similar to 'EC', the term 'SEB', which I guess is the surface energy balance, also has the similar problem. It first appears in Line 252, but surface energy balance first appears in Line 151. And please also check the upper/lower case together. For example, in Line 151, it appears that the sentence should be: The surface energy balance (SEB) and AL-internal heat fluxes towards......

We agree and have modified the manuscript accordingly.

2. In line 269, QLE should refer to latent rather than sensible heat flux.

Yes, thank you for spotting this error. The sentence should read: "The evaporative water flux is derived from the latent turbulent flux QLE..."

3. There is no need to add a subheading in line 361 (4.2.1 Field observations). The paragraph can directly follow line 360.

We agree and have modified the manuscript accordingly.

4. Some sentences are repeated, please check. For example, the sentence in the caption of Figure 2 is mentioned again in lines 349-350.

We have removed repeated sentences for brevity.

REFERENCES

Amschwand, D., Scherler, M., Hoelzle, M., Krummenacher, B., Haberkorn, A., Kienholz, C., and Gubler, H.: Surface heat fluxes at coarse blocky Murtèl rock glacier (Engadine, eastern Swiss Alps), The Cryosphere Discussion, https://doi.org/10.5194/egusphere-2023-2109, 2023. (This publication has been accepted in the meantime)

Amschwand, D.,Wicky, J., Scherler, M., Hoelzle,M., Krummenacher, B., Haberkorn, A., Kienholz, C., and Gubler, H.: Sub-surface processes and heat fluxes at coarse-blocky Murtèl rock glacier (Engadine, eastern Swiss Alps), in review at Earth Surface Dynamics, 2024.

Haeberli, W., ed.: Pilot analysis of permafrost cores from the active rock glacier Murtèl I, Piz Corvatsch, Eastern Swiss Alps. A workshop report., no. 9 in Arbeitsheft, VAW/ETH Zürich, 1990.

Jones, D. B., Harrison, S., Anderson, K., and Whalley,W. B.: Rock glaciers and mountain hydrology: A review, Earth-Science Reviews, 193, 66–90, https://doi.org/10.1016/j.earscirev.2019.04.001, 2019.

Louis, C., Halloran, L. J. S., and Roques, C.: Seasonal and diurnal freeze-thaw dynamics of a rock glacier and their impacts on mixing and solute transport, EGUsphere [preprint], https://doi.org/10.5194/egusphere-2024-927, 2024.

Stucki, T.: Permafrosttemperaturen im Oberengadin. Eine Auswertung der Bohrlochtemperaturen im alpinen Permafrost des Oberengadins im Hinblick auf einen Erwärmungstrend und Schmelzwasserabfluss aus dem Permafrost., Master's thesis, Abteilung Erdwissenschaften der ETH Zürich, 1995.

Tenthorey, G. and Gerber, E.: Hydrologie du glacier rocheux de Murtèl (Grisons). Description et interprétation de traçages d'eau, in: Modèles en Géomorphologie – exemples Suisses. Session scientfique de la Société suisse de Géomorphologie. Fribourg, 22/23 juin 1990., edited

by Monbaron, M. and Haeberli, W., vol. 3 of Rapport et recherches, Institut de Géographie Fribourg, 1991.

---

## Author Response (AR2)

Dear editor Hongkai Gao, Dear reviewers

We thank you again for the constructive and careful reviews. We have modified the manuscript according to the reviewers' requests and provide detailed answers to their issues below. Here, we summarize the most important changes:

- Added two figures (Figs. 3, 11) containing field photos of the measurement stake (as requested by referee #3)
- We made minor changes to the conclusions (as requested by referee #3) to increase the clarity of the message.

We hope that we could satisfactorily address the remaining issues.

Kind regards

Dominik Amschwand on behalf of all co-authors

In the following, we would like to reply to their comments from Anonymous referee #3 in detail (reviewer comment in blue, >our response in black):

General comments:
The manuscript "Seasonal ice storage changes and meltwater generation at Murtèl rock glacier (Engadine, eastern Swiss Alps): Estimates from measurements and energy budgets in the coarse blocky active layer" by Amschwand et al. presents a complete hydro-meteorological dataset to investigate the processes of the rock glacier store-release at Swiss Alps. Based on below-ground energy-flux measurements, ground-ice melt and hydrological measurements, the authors conclude that the rock glacier plays a crucial role in buffering seasonal runoff through active layer ice melt and thermal buffering. The active layer acts as a buffer, refreezing snowmelt in the winter and slowly releasing meltwater throughout the thaw season, which helps maintain baseflow. The findings highlight the importance of rock glaciers in regulating water flow. My major concerns are as follows:

1. Some key details are missing, especially the ablation measurements of the ground ice. Since the evolution of the ground-ice is an important part of this study, how to observe it determines whether the conclusions of the paper are reliable. It would be beneficial to provide supporting photos. >Yes, we agree and added two figures containing photos of the measurement stake.
2. Two or three years of observation is relatively short and should not be considered as long-term changes. The term 'long-term' is mentioned only twice in the entire paper. The first time is in the introduction of the article's research content, and the second time is in the conclusion. >Yes, we agree and deleted 'long-term'.
3. In this study, the focus seems to be on measuring direct runoff during summer. Is there surface runoff in the watershed in spring? Is some of the winter precipitation melting in the spring and forming runoff? >Yes, field observations and the discharge measurement show that there is surface runoff of snowmelt. We added the underlined part to the sentence in the conclusion to clarify that not all snowmelt is converted to ground ice: "A substantial part of the snowmelt is rapidly fixed as AL ice in winter--spring, but slowly released during the thaw season..."

Specific Comments:

Line 19: This expression (a few%) looks weird to me. >We have spelled out the "%" sign in words.

Line 43: cf. >We have deleted the cf and the unnecessary reference.

Line 47-49: Additional references needed. >We have added references.

Line 116: There is no need to cite this article here. >Yes, we have deleted it.

Line 124: the permafrost-groundwater connectivity. As far as I understand, the whole article is about the relationship between permafrost and surface runoff. > Yes indeed, and we also put forward the hypothesis that rock glaciers contribute to groundwater recharge. We extended the sentence (additions underlined): "This case study contributes to the question of the hydrological significance of rock glaciers by presenting a complete hydro-meteorological data set at the well-studied Murtèl rock glacier, investigates how the permafrost affects the surface runoff pattern, and explores implications for the permafrost–groundwater connectivity."

Line 184: What does the question mark here mean? >We replaced "intra-permafrost?" by "likely intra-permafrost".

Line 195-Line 199: Please give the full forms of these abbreviations >We spelled out "Swiss Permafrost Monitoring Network" (PERMOS). PERMA-XT is the project name.

Section 3.4.1: Adding a photo or two from the observations would make this more convincing. >Yes, we have added photographs.

Figure 7: A small rainfall in August 2022 produced a large runoff, exceeding the runoff of many large rainfalls in 2021(even > 25mm), and an explanation should be added. >The reason for the "excess" streamflow at relatively low rainfall is that this discharge compounded with discharge of previous rainfalls. We added an explanation to the figure caption: "Note that discharge is in cases compounded with discharge from previous rainfalls, introducing some scatter in the rainfall–streamflow relation".

Line 675: old permafrost ice on long-term (over millennia). This is not the result of this article's research. It may have melted in decades of years. >Yes, we agree. We deleted the sentences because this is already explained in the introduction:  We quantify supra-permafrost ice storage changes…".